# Intrinsic timescales as an organizational principle of neural processing across the whole rhesus macaque brain

**Ana MG Manea[1,2]\*, Anna Zilverstand[3,4], Kamil Ugurbil[2,5,6], Sarah R Heilbronner[1,4,5], Jan Zimmermann[1,2,4,5,7]**

[1]Department of Neuroscience, University of Minnesota, Minneapolis, United States; [2]Center for Magnetic Resonance Research, University of Minnesota, Minneapolis, United States; [3]Department of Psychiatry & Behavioral Sciences , University of Minnesota, Minneapolis, United States; [4]Medical Discovery Team on Addiction, University of Minnesota, Minneapolis, United States; [5]Center for Neuroengineering, University of Minnesota, Minneapolis, United States; [6]Department of Radiology, University of Minnesota, Minneapolis, United States; [7]Department of Biomedical Engineering, University of Minnesota, Minneapolis, United States

**\*For correspondence:**
manea006@umn.edu

**Competing interest:** The authors declare that no competing interests exist.

**Abstract** Hierarchical temporal dynamics are a fundamental computational property of the brain; however, there are no whole brain, noninvasive investigations into timescales of neural processing in animal models. To that end, we used the spatial resolution and sensitivity of ultrahigh field functional magnetic resonance imaging (fMRI) performed at 10.5 T to probe timescales across the whole macaque brain. We uncovered within-species consistency between timescales estimated from fMRI and electrophysiology. Crucially, we extended existing electrophysiological hierarchies to whole-brain topographies. Our results validate the complementary use of hemodynamic and electrophysiological intrinsic timescales, establishing a basis for future translational work. Further, with these results in hand, we were able to show that one facet of the high-dimensional functional connectivity (FC) topography of any region in the brain is closely related to hierarchical temporal dynamics. We demonstrated that intrinsic timescales are organized along spatial gradients that closely match FC gradient topographies across the whole brain. We conclude that intrinsic timescales are a unifying organizational principle of neural processing across the whole brain.

## Editor's evaluation

Neural activity measured in both electrophysiological and functional neuroimaging experiments are often temporally correlated, and the timescales of such correlation in ongoing neural activity, or intrinsic neural timescales, show a hierarchical pattern across the cortical surface. The present study establishes a close link between these timescales and functional connectivity in the brains of non-human primates, suggesting that temporal autocorrelation is an important organizing feature of large-scale neural activity.

## Introduction

Recent work supports the idea that the neural processing is organized over multiple timescales (*Cavanagh et al., 2020*; *Hasson et al., 2008*; *Honey et al., 2012*; *Kiebel et al., 2008*; *Murray et al., 2014*; *Raut et al., 2020*; *Soltani et al., 2021*). Hierarchical temporal dynamics are a fundamental computational property of the brain—fast-changing sensory input is encoded by the fast dynamics

of sensory areas, which is in turn integrated into temporally extended representations encoded by the slower dynamics of higher-order areas (*Cavanagh et al., 2020*; *Raut et al., 2020*; *Soltani et al., 2021*). Temporal integration and information processing over multiple timescales are crucial for understanding how the brain deals with the complexity of the real world (*Zimmermann et al., 2018*); hence, understanding whole-brain temporal dynamics is highly important.

As one moves from sensory to associative areas, neural processing reflects a hierarchy of progressively longer 'temporal receptive windows' (TRWs)—defined as the length of time during which an area accumulates information during task performance (*Hasson et al., 2008*). In other words, the range of the TRW of each area reflects its functional role, with sensory areas having short TRWs, to enable fast sensory processing, and associative areas having longer TRWs, allowing them to process information unfolding over longer periods of time. Not only does neural processing display these timescales during task performance, but we now know that temporal specialization arises from a region's intrinsic dynamics, that is, temporal dynamics that are present even in the absence of direct sensory stimulation. This was first described by *Murray et al., 2014* analyzing action potentials of single neurons in the macaque. They showed that timescales of baseline neuronal activity vary across cortical areas, revealing a hierarchical progression from fast timescales in sensory areas, to slower timescales in association areas. These intrinsic fluctuations in the neural signal, also described as intrinsic neural timescales (INTs), are thought to set the duration over which brain regions integrate inputs (*Murray et al., 2014*). They are hypothesized to be the result of multiple neuronal mechanisms, including local dynamics and recurrent connectivity (*Murray et al., 2014*; *Soltani et al., 2021*; *Watanabe et al., 2019*; *Demirtaş et al., 2019*).

Additional studies have replicated and extended the INT hierarchy, reinforcing the idea that brain regions exhibit unique timescales in their intrinsic activity, which reflect functional specialization in terms of temporal integration (*Cavanagh et al., 2016*; *Cirillo et al., 2018*; *Fascianelli et al., 2019*; *Murray et al., 2014*; *Ogawa and Komatsu, 2010*; *Rossi-Pool et al., 2021*; *Maisson et al., 2021*). The concept of INT hierarchy has even been extended to subcortical regions—specifically the basal ganglia (*Nougaret et al., 2021*). Moreover, a large-scale network model, based on anatomical connectivity (AC) and the strength of local recurrent connectivity, has been used to successfully reproduce the temporal hierarchy (*Chaudhuri et al., 2015*). This suggests that AC is closely related to INTs.

Alterations in hierarchical INT patterns can result in cognitive and behavioral abnormalities (*Watanabe et al., 2019*; *Wengler et al., 2020*; *Zilio et al., 2021*). For example, abnormal INT patterns have been associated with core symptoms of high-functioning individuals with Autism Spectrum Disorder (*Watanabe et al., 2019*). Likewise, delusions and hallucinations are associated with INT alterations at different hierarchical levels, leading to symptom-specific changes in hierarchical INT gradients (*Wengler et al., 2020*). Hence, investigations into INTs have the potential to clarify hierarchical dysfunction in whole-brain dynamics in neuropsychiatric disorders.

Traditionally, INTs have been investigated using invasive electrophysiology, which is not suited to uncover whole-brain dynamics. It is also challenging to collect in humans, particularly for large data sets (as would be needed to assess differences across populations). One way to overcome this problem is using functional magnetic resonance imaging (fMRI), which offers whole-brain coverage and can be performed across species and in larger sample sizes. However, it is not yet clear whether hemodynamic INTs derived from fMRI reflect similar neural phenomena as those derived from electrophysiology. To clarify this issue, electrophysiological and hemodynamic INTs must be compared directly within the same species. To that end, recent advancements in nonhuman primate (NHP) neuroimaging, based on the use of very high magnetic fields (10.5 T) for significantly boosting signal-to-noise ratios (SNRs) (*Lagore et al., 2021*; *Yacoub et al., 2020*), make it possible to study hemodynamic INTs in the macaque brain. Moreover, taking advantage of the whole-brain coverage of fMRI, we are investigating INT topography. This is important since more recent work has shown that there is significant heterogeneity within regions (*Cavanagh et al., 2020*; *Cirillo et al., 2018*; *Nougaret et al., 2021*), even though initially INTs were assigned to cortical regions as a whole.

In addition to INTs, gradients of functional connectivity (FC) can also describe the functional architecture of neural processing (*Haak et al., 2018*; *Vos de Wael et al., 2020*). Gradient approaches map macroscale brain features to low-dimensional manifold representations (*Vos de Wael et al., 2020*). Gradient mapping has the potential to uncover overlapping functional topographies by treating the high-dimensional connectivity structure as a spatial variance decomposition problem. Notably, this

approach finds patterns in the high-dimensional FC similarity structure that describe the spatial direction of maximum FC variance. For example, areas in the early visual cortex are known to exhibit at least two gradual modes of functional change (i.e., eccentricity and polar angle maps), which can be successfully retrieved from resting-state FC via gradient mapping (*Haak et al., 2018*). The existence of multiple overlapping principles of topographic organization is likely a general property of the whole brain (*Haak et al., 2018*; *Harvey et al., 2020*; *Jbabdi et al., 2013*; *Marquand et al., 2017*; *Przezdzik et al., 2019*). Here, we carry out an investigation into the multidimensional functional architecture of the macaque brain by probing its spatial and temporal intrinsic dynamics. This allows us to achieve several significant goals. First, we are able to relate fMRI hemodynamic timescales to the gold standards of electrophysiology within the same species. Second, with these results in hand, we are able to show that one facet of the high-dimensional FC topography of any region in the brain is closely related to INT topography. We show that the hierarchical intrinsic dynamics of neural processing manifest into gradients of INTs and FC which follow the same gradual topographical arrangement. In sum, we reliably describe one facet of the macroscale functional organization of the brain, with respect to spatial and temporal dynamics.

## Results
### INT hierarchies

First, we set out to replicate and validate findings from electrophysiological INT estimation with ultra-high field fMRI. *Murray et al., 2014* initially described (in macaques) a visual cortical hierarchy of seven cortical areas: middle temporal (MT) area, lateral intraparietal (LIP) area, orbitofrontal cortex (OFC), lateral PFC (LPFC), anterior cingulate cortex (ACC); and a somatosensory hierarchy including primary somatosensory (S1) and secondary somatosensory (S2) cortex (*Figure 1A–B*, left). This cortical hierarchy has been replicated and further extended by assigning INTs to the dorsal premotor (PMd) cortex, frontopolar cortex (PFp), ventral and dorsal LPFC (VLPFC and DLPFC) (*Cavanagh et al., 2016*; *Cirillo et al., 2018*; *Fascianelli et al., 2019*; *Figure 1C–D*, left). Recently, INTs have been estimated in the medial wall of the PFC, including the ventromedial PFC (vmPFC), subgenual ACC (sgACC), pregenual ACC (pgACC), and dorsal ACC (dACC) (*Maisson et al., 2021*; *Figure 1E*, left). Finally, a subcortical INT hierarchy has been described for regions of the basal ganglia: external segment of the globus pallidus (GPe), subthalamic nucleus (STN), and the striatum (*Nougaret et al., 2021*; *Figure 1F*, left).

The electrophysiology-based (*Figure 1*, left) and fMRI-based (*Figure 1*, right) INTs reveal similar hierarchies across the brain. By replicating these physiological hierarchies, we demonstrate that fMRI is well-suited to investigate aspects of whole-brain temporal dynamics. Minor deviations from hierarchies estimated from single-unit recordings (e.g., in STN, sgACC, and pgACC) can be attributed to several factors such as a high degree of INT heterogeneity within these regions, and differences between studies in the atlases used and size of the areas under investigation (see *Figure 1—figure supplement 3* for the INT stability analysis).

Our definition of INTs combines the effects of ACF magnitude and timescale (i.e., the 0-crossing lag). Since it has previously been shown that the magnitude of the ACF values, parameterized as the lag-1 ACF value, differs across brain areas (*Shinn et al., 2021*), we further examined whether the observed hierarchies are driven by magnitude, timescale, or both. In line with previous results, the lag-1 ACF values systematically varied across the brain (*Shinn et al., 2021*). Moreover, the lag-1 ACF values and the 0-crossing lag were linearly related to INTs as operationalized in this study (see *Figure 1—figure supplement 1* for voxel-wise whole-brain maps of the lag-1, 0-crossing lag, and INTs). The maps suggest that magnitude and timescale are closely related. To exclude the possibility that the observed hierarchies are driven by magnitude, rather than timescale, we re-computed them by using the 0-crossing lag as a proxy for INT. To obtain a time estimate of hemodynamic timescales, the 0-lag crossing lag was multiplied by the TR (i.e., 1.1 s). The hierarchical ordering of both cortical and subcortical areas is consistent (see *Figure 1—figure supplement 2*) with that based on our INT measure (see *Figure 1*). Hence, while the ACF magnitude is closely related to our INT index and the 0-crossing lag, temporal hierarchies are not driven by heterogeneity in the ACF magnitude.

We next investigated previously unexamined regions, making unique use of our high resolution, whole-brain coverage. Specifically, we reveal a hierarchy of INTs in the dorsal visual pathway (*Goodale*

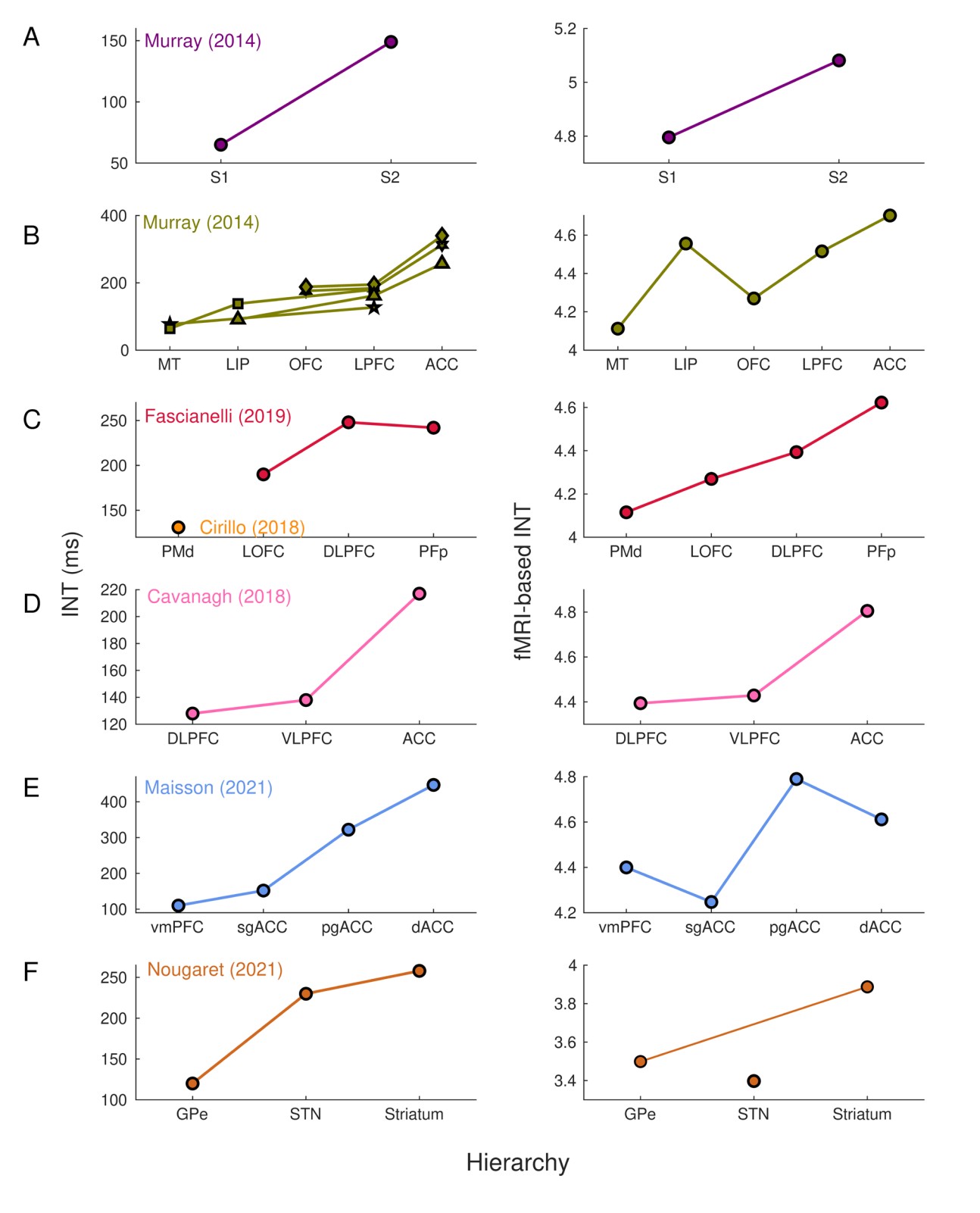

**Figure 1.** Intrinsic neural timescales estimated from functional magnetic resonance imaging (fMRI) and electrophysiology reveal a similar hierarchical ordering of cortical and subcortical areas. Group-averaged (N=9) INTs were estimated for each area of interest by averaging across voxels. Each color represents a hierarchy estimated from a different electrophysiology data set (Left). The hierarchies estimated from spiking data were replicated via fMRI (Right). Somatosensory (A) and visual (B) hierarchies reported by *Murray et al., 2014*. Note: Each symbol represents a different data set as originally

*Figure 1 continued*

reported by the respective authors. Frontal hierarchy reported by *Fascianelli et al., 2019* and *Cirillo et al., 2018* (C). Frontal hierarchy reported by *Cavanagh et al., 2016* (D). Medial prefrontal hierarchy reported by *Maisson et al., 2021* (E). Subcortical hierarchy reported by *Nougaret et al., 2021* (F). The areas were defined bilaterally for each panel to match the recording sites of the data set—the Cortical Hierarchy Atlas of the Rhesus Macaque (*Jung et al., 2021*) and Subcortical Atlas of the Rhesus Macaque (*Hartig et al., 2021*) were used when possible (see *Supplementary file 1*). The hemodynamic INTs were estimated as the sum of the autocorrelation function (ACF) values in the initial positive period—that is, this measurement considers both the number of lags and the magnitude of the ACF values. Hence, the INT does not have a time unit and it is used as a proxy of the timescale of that area, with higher values reflecting longer timescales. See *Figure 1—figure supplement 2* for an estimate of the time constant of each area. The INT of individual areas was estimated by taking the average of the voxels' INT values. The analysis was performed at the group level (N=9). Abbreviations: S1/S2 (primary/secondary somatosensory cortex), MT (middle temporal area), LIP (lateral intraparietal area), OFC (orbitofrontal cortex), LPFC (lateral prefrontal cortex), ACC (anterior cingulate cortex), PMd (dorsal premotor cortex), LOFC (lateral OFC), DLPFC (dorso-lateral PFC), VLPFC (ventro-lateral PFC), PFp (polar prefrontal cortex), vmPFC (ventro-medial PFC), sgACC (subgenual ACC), pgACC (pregenual ACC), dACC (dorsal ACC), GPe (external globus pallidus), STN (subthalamic nucleus). *Figure 1—figure supplement 3* depicts single-subject INT stability and similarity and the group-level stability of the INT hierarchies.

The online version of this article includes the following figure supplement(s) for figure 1:

**Figure supplement 1.** Whole-brain hemodynamic intrinsic neural timescales (INTs), lag-1 and 0-crossing lag.

**Figure supplement 2.** The time constants of hemodynamic intrinsic neural timescales (INTs).

**Figure supplement 3.** Stability and similarity analysis of whole-brain intrinsic neural timescales (INTs).

*et al., 1994*; *Goodale and Milner, 1992*; *Figure 2*). INTs gradually increase from the entry point of visual information to higher-order areas involved in visuo-motor integration (*Figure 2*, inset). INTs are hierarchically organized within the occipito-parietal, parieto-premotor, and parieto-prefrontal components of the dorsal pathway (*Kravitz et al., 2011*). Also, in line with the occipito-parietal pathway being the source of the other two pathways, it is lower in the hierarchy. Taken together, we replicate electrophysiological INT hierarchies and demonstrate the invaluable contribution of using whole-brain fMRI to extend this research by describing a previously uncharacterized fine-grained temporal hierarchy along the visual pathway as an example.

## INT hierarchies are closely related to FC gradients across the brain

We next examined the relationship between INTs and gradients of FC at different levels of description, starting with the frontal and parietal lobes (see *Figure 3*, gradients computed twice, once using within-lobe FC and again using frontal-parietal FC). In the frontal lobe, the first component (i.e., G1) shows an overall anterior-posterior organization spanning from sensory to motor areas. This aligns with previous studies showing a global anterior-posterior functional gradient in the frontal lobe (*Badre, 2008*; *Riley et al., 2018*; *Tan et al., 2020*). The second component (i.e., G2) shows an overall lateral-medial organization, extending from the PMv to the medial wall. The INT map was correlated with G2 for both within-area (r=0.47) and fronto-parietal (r=–0.26) connectivity. In the parietal lobe, G1 showed a medial-lateral and G2 a ventro-dorsal overall organization. The INT map was correlated with G1 for both within-area (r=0.56) and fronto-parietal (r=0.47) connectivity. (Note that the sign of the gradients is arbitrary as a byproduct of the decomposition method [*Haak and Beckmann, 2020*]; hence, only the magnitude but not the direction of the INT-FC gradient correlation is relevant.)

We then selected functionally meaningful subdivisions of the frontal lobe to probe finer-grained connectivity profiles and interrogate the existence of multiple hierarchical gradients of function (see *Figure 4*). In all cases, the INT maps were highly correlated with one of the principal components estimated from within-area connectivity (i.e., LPFC G1: r=0.35; Motor G1: r=0.68; OFC G2: r=0.45; MPFC G1: r=0.49). For example, the motor cortex has a well-established hierarchy, with premotor regions at the top and M1 at the bottom (*Binkofski and Buxbaum, 2013*; *Wong et al., 2015*). Accordingly, the maps reflected this functional hierarchy, with M1 being at one end and the PMv at the other. The OFC shows a predominantly anterior-posterior axis of change, which agrees with anatomical findings—the caudal and lateral regions are more input-like unimodal regions while more anterior areas integrate these multimodal signals (*Carmichael and Price, 1994*; *Carmichael and Price, 1996*; *Ongür and Price, 2000*; *Price, 1999*). The lateral and medial subdivisions of the PFC revealed a predominately anterior-posterior organization. The same pattern of results was observed at the single-subject level (see *Figure 4—figure supplement 2*).

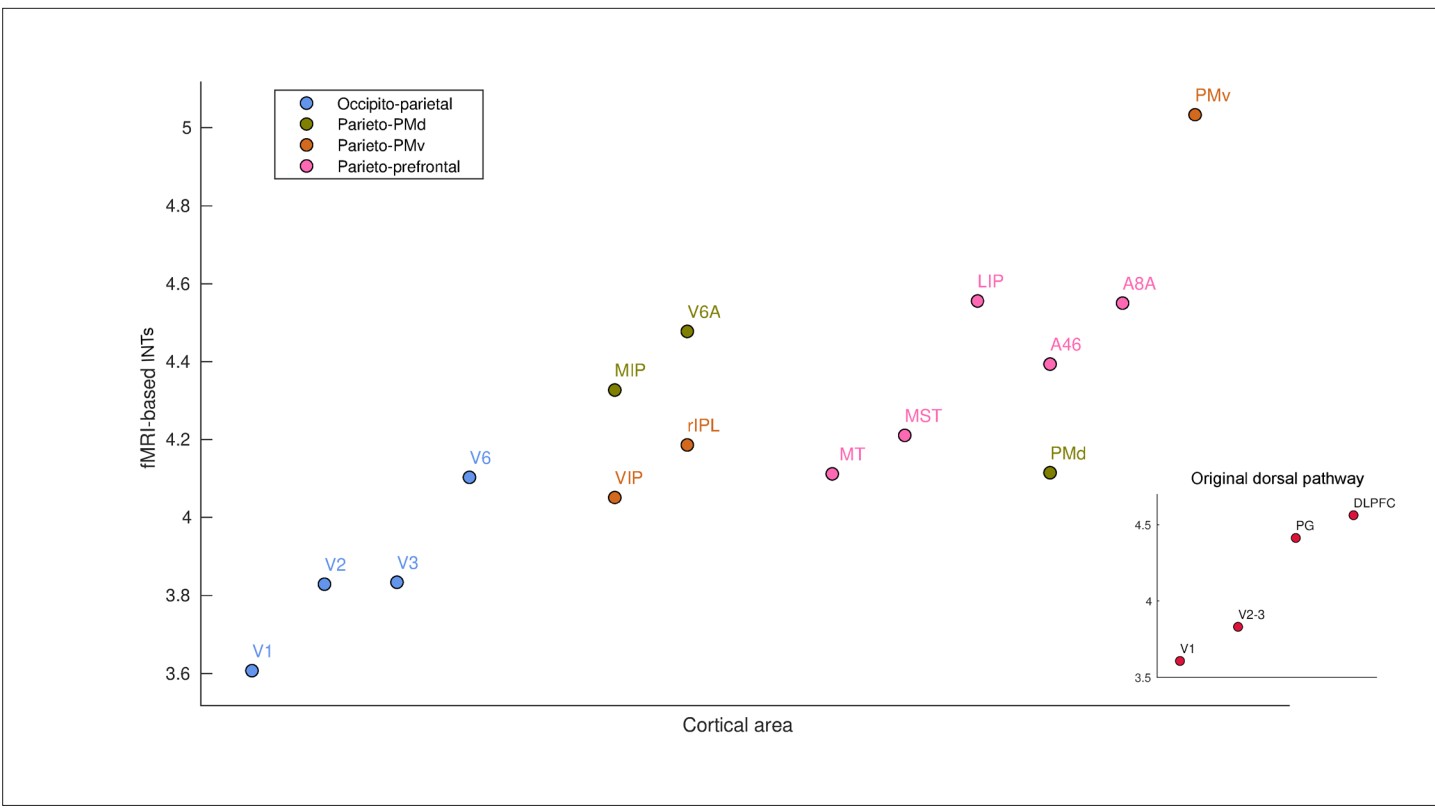

**Figure 2.** Functional magnetic resonance imaging (fMRI) reveals a hierarchical ordering of timescales in the dorsal pathway for visuospatial processing. Group-averaged (N=9) intrinsic neural timescales (INTs) were estimated for each area of interest by averaging across voxels. The areas were defined bilaterally based on the Cortical Hierarchy Atlas of the Rhesus Macaque (*Jung et al., 2021*) (see *Supplementary file 2*). The dorsal pathway reveals a gradual progression of INTs from sensory to association and motor cortices. The hierarchical pattern of INTs can be observed in the original description of the dorsal pathway (inset), but also in more recently described subdivisions. Every color represents a subdivision of the dorsal pathway as described in *Kravitz et al., 2011*. Note: Since the hemodynamic INTs were estimated as the sum of ACF values in the initial positive period, this measurement considers both the number of lags and the magnitude of the ACF values. As a result, the INT measure does not have a time unit and it is used as a proxy of the timescale of that area, with larger values reflecting longer timescales. Abbreviations: DLPFC (dorso-lateral prefrontal cortex), VIP (ventral intraparietal area), MIP (medial intraparietal area), rIPL (rostral inferior parietal lobule), MT (middle temporal area), MST (medial superior temporal area), LIP (lateral intraparietal area), A46 (area 46), A8A (area 8A), PMd (dorsal premotor cortex), PMv (ventral premotor cortex).

Although we were mainly interested in the fine-grained functional organization of individual areas, we also asked whether the INT-FC relationship holds at the whole-cortex level (see *Figure 3—figure supplement 1*). It is important to note that the whole-cortex estimation reflects overall inter-areal FC differences rather than the fine-grained FC patterns in smaller areas; that is because inter-areal connectivity differences are more pronounced than intra-areal differences. The whole-cortex INT map was highly correlated with the second gradient estimated from the cortical FC matrix (r=0.70). With these results in hand, we conclude that INTs and FC gradients are closely related and represent a functional principle of organization across the whole brain.

We then investigated cortical-subcortical hierarchical gradients of function. Cortico-striatal AC has been extensively studied in NHPs, however, striatal-cortical FC and INTs have not. The striatal INT map was correlated with the FC gradient estimated from cortico-striatal (r=0.38) connectivity (see *Figure 5*). The cortico-striatal FC topography exhibited a gradual pattern of connectivity change corresponding to the anterior-posterior, dorsal-ventral, and medial-lateral anatomical gradient of connectivity derived from tract-tracing studies in NHPs (*Haber, 2016*; *Haber and Knutson, 2010*; *Friedman et al., 2002*; *Alexander et al., 1986*). INTs followed a similar pattern. Moreover, the FC gradients were similar to those estimated from human neuroimaging data (*Marquand et al., 2017*). To examine the underlying FC patterns, we projected the striatal gradient onto the cortex (see *Figure 5A*). This analysis not only validated the meaning of the gradients, but also revealed that the cortico-striatal FC closely matched AC. To further investigate the link between FC and INTs, we

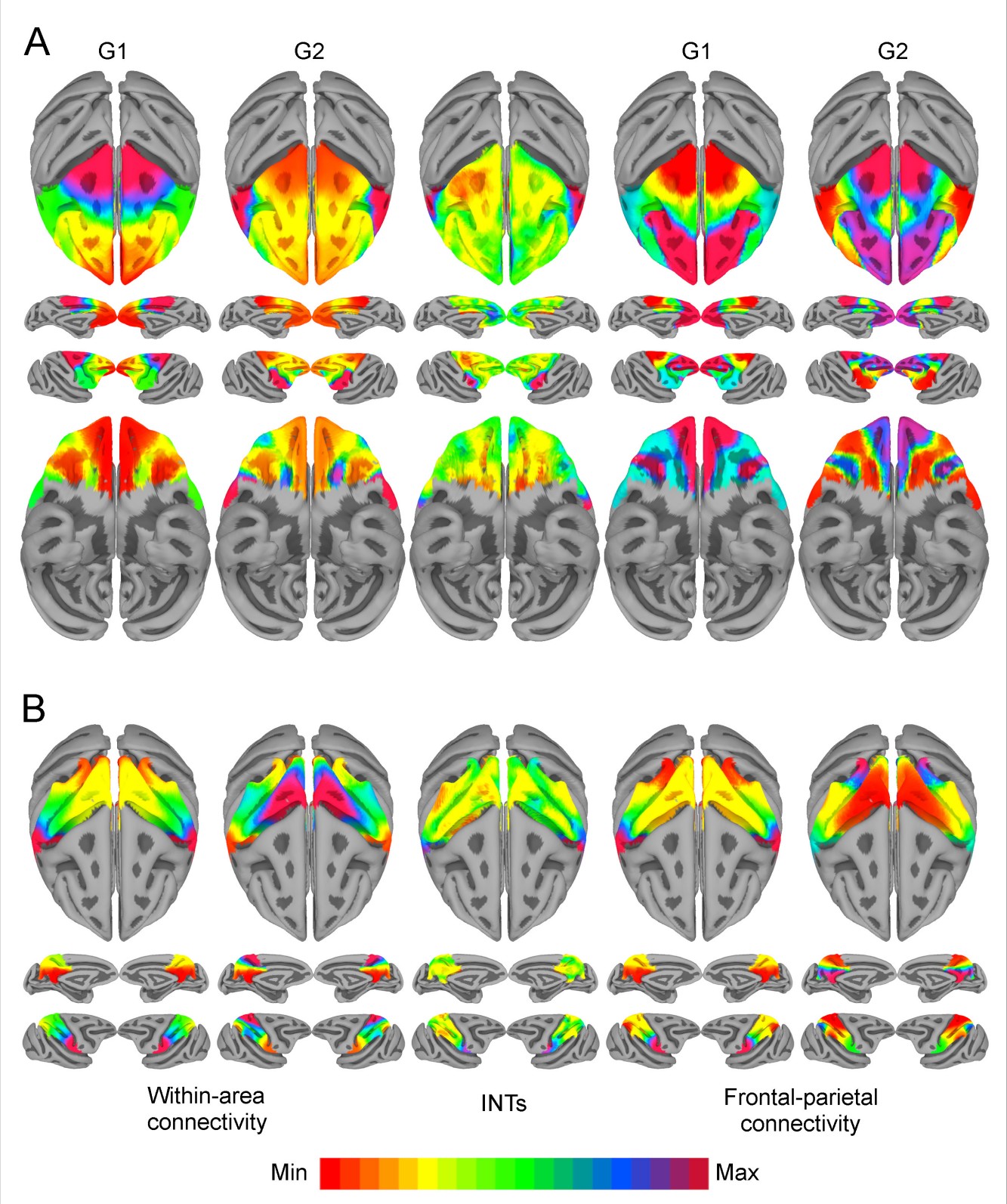

**Figure 3.** Functional connectivity gradients are related to intrinsic neural timescale (INT) topographies. INT maps (Middle) and functional connectivity (FC) gradients were computed at the group level (N=9). Gradient 1 (G1) and Gradient 2 (G2) were estimated using a cosine similarity affinity computation of the within-area FC (Left) and frontal-parietal FC (Right), followed by diffusion mapping (Note: G1 and G2 are the first components that describe the axes of largest variance). (A) Frontal INTs were correlated with G2 for both within-area (r=0.47) and frontal-parietal connectivity (r=–0.26). (B)

*Figure 3 continued on next page*

*Figure 3 continued*

Parietal INTs were correlated with G1 for both within-area (r=0.56) and parietal-frontal connectivity (r=0.47). The color bar indicates the position along the FC gradient (Note: values are on an arbitrary scale) or INT values (Note: since the hemodynamic INTs were estimated as the sum of ACF values in the initial positive period, this measurement considers both the number of lags and the magnitude of the ACF values. As a result, the INT does not have a time unit and it is used as a proxy of the timescale of that area, with larger values reflecting longer timescales). The value ranges differ across panels unless otherwise stated.

The online version of this article includes the following figure supplement(s) for figure 3:

**Figure supplement 1.** Intrinsic neural timescales (INTs) are topographically organized along functional connectivity (FC) gradients across the whole cortex.

reconstructed the striatal INT map by replacing the INT of every voxel with the average INT value of the cortical voxels (i.e., 10 voxels) with which they are most correlated (see *Figure 5B*). In accordance with the above-mentioned results, the striatal INTs and reconstructed cortico-striatal INT maps were correlated (r=0.45). At the voxel level, the INT values closely matched the average INTs of their top 10 strongest cortical connections.

Overall, we demonstrated that INTs are organized along spatial gradients that closely match FC gradients.

## Discussion

Temporal integration is a fundamental property of neural processing (*Cavanagh et al., 2016*; *Hasson et al., 2008*; *Honey et al., 2012*; *Soltani et al., 2021*), both during task performance and in the absence of external stimulation (*Cirillo et al., 2018*; *Hasson et al., 2008*; *Murray et al., 2014*; *Raut et al., 2020*; *Ito et al., 2020*). However, until this point, there have been major gaps in investigating temporal integration across the brain, across species, and across populations. This is because, traditionally, temporal integration has been studied primarily using single-unit electrophysiology in NHPs (see *Gao et al., 2020*). Although there is some evidence that BOLD signals from fMRI may allow for whole-brain interrogation of timescales in humans (*Watanabe et al., 2019*; *Wengler et al., 2020*), such an approach had not yet been validated. Here, we focused on investigating whole-brain INT and FC topographies at different levels of description, to first validate, within-species, the study of INTs using BOLD fMRI, and then investigate the relationship between temporal and spatial dynamics of the resting brain with ultrahigh resolution fMRI across the whole macaque brain.

We have demonstrated the feasibility of estimating INTs from fMRI in the NHP brain. We did so by showing that hemodynamic INT hierarchies closely match those derived from NHP electrophysiology. While some recent human neuroimaging studies have investigated whole-brain hemodynamic INTs (*Raut et al., 2020*; *Watanabe et al., 2019*; *Wengler et al., 2020*), temporal hierarchies derived from electrophysiology have not been replicated via fMRI, nor within species. Furthermore, in a previous paper, INTs derived from fMRI data in human subjects were elegantly validated with EEG (*Watanabe et al., 2019*). The present findings are in agreement with these results, finding slower and faster INTs in frontoparietal areas and sensory-related areas, respectively. This points toward cross-species homologies in cortical INTs, hence, laying the groundwork for future translational work. Nevertheless, the step presented here is essential, because we were able to demonstrate the equivalence of the fMRI estimates of INT hierarchies with single-neuron spiking results within the same species. This finding provides validation for the complementary use of hemodynamic and electrophysiological INTs. Using fMRI in this context has several advantages. First, it can be performed on the whole brain. Second, it can easily be employed across different human populations in the future, dramatically expanding the possible uses of INT investigations.

We did observe minor deviations from electrophysiological INTs in our data. These may be in large part attributable to a high degree of INT heterogeneity within individual anatomical regions. This observation is in line with previous reports of heterogeneity within neural populations in single-unit recordings (*Cirillo et al., 2018*; *Nougaret et al., 2021*). This suggests that averaging INTs within anatomical regions might not provide a full description of macroscale organization.

It is worth noting that the current study could not resolve the exact correspondence between the time constants estimated from spiking activity and BOLD. The relationship between neural activity and the BOLD signal has been revealed via simultaneous electrophysiological-fMRI recording—that

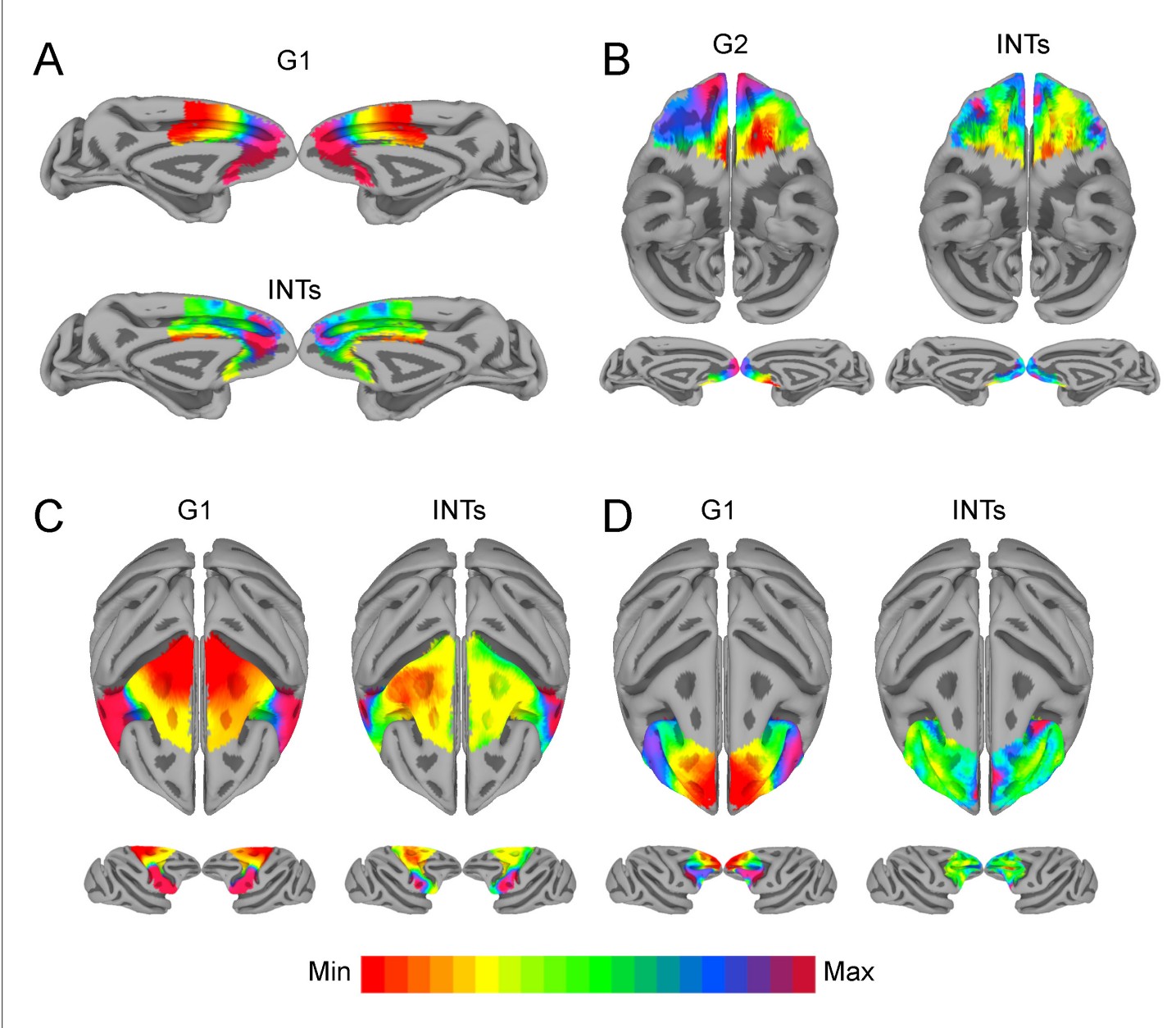

**Figure 4.** The relationship between functional connectivity (FC) and intrinsic neural timescales (INTs) in the frontal lobe. Group-averaged INT maps (Right) and FC gradients (Left) were estimated at the group level (N=9). Gradient 1 (G1) and Gradient 2 (G2) were estimated using a cosine similarity affinity computation of the within-area followed by diffusion mapping (Note: G1 and G2 are the first components that describe the axes of largest variance). The FC gradient with the highest correlation to INTs was plotted. (**A**) INTs in medial prefrontal cortex were correlated with G1 (r=0.49). (**B**) INTs in the orbitofrontal cortex were correlated with G2 (r=0.45). (**C**) INTs in the motor cortex were correlated with G1 (r=0.68). (**D**) INTs in the lateral prefrontal cortex were correlated with G1 (r=0.35). The color bar indicates the position along the FC gradient (values are on an arbitrary scale) or INT values (Note: since the hemodynamic INTs were estimated as the sum of ACF values in the initial positive period, this measurement considers both the number of lags and the magnitude of the ACF values. As a result, the INT does not have a time unit and it is used as a proxy of the timescale of that area, with larger values reflecting longer timescales). The value ranges differ across panels unless otherwise stated. For a more in-depth analysis of the relationship between INTs and FC, see *Figure 4—figure supplement 1*. For the single-subject analysis, see *Figure 4—figure supplement 2*.

The online version of this article includes the following figure supplement(s) for figure 4:

**Figure supplement 1.** Intrinsic neural timescales (INTs) and functional connectivity (FC) gradients are monotonically related in the frontal lobe.

**Figure supplement 2.** Intrinsic neural timescales (INTs) closely match functional connectivity (FC) gradients at the single-subject level.

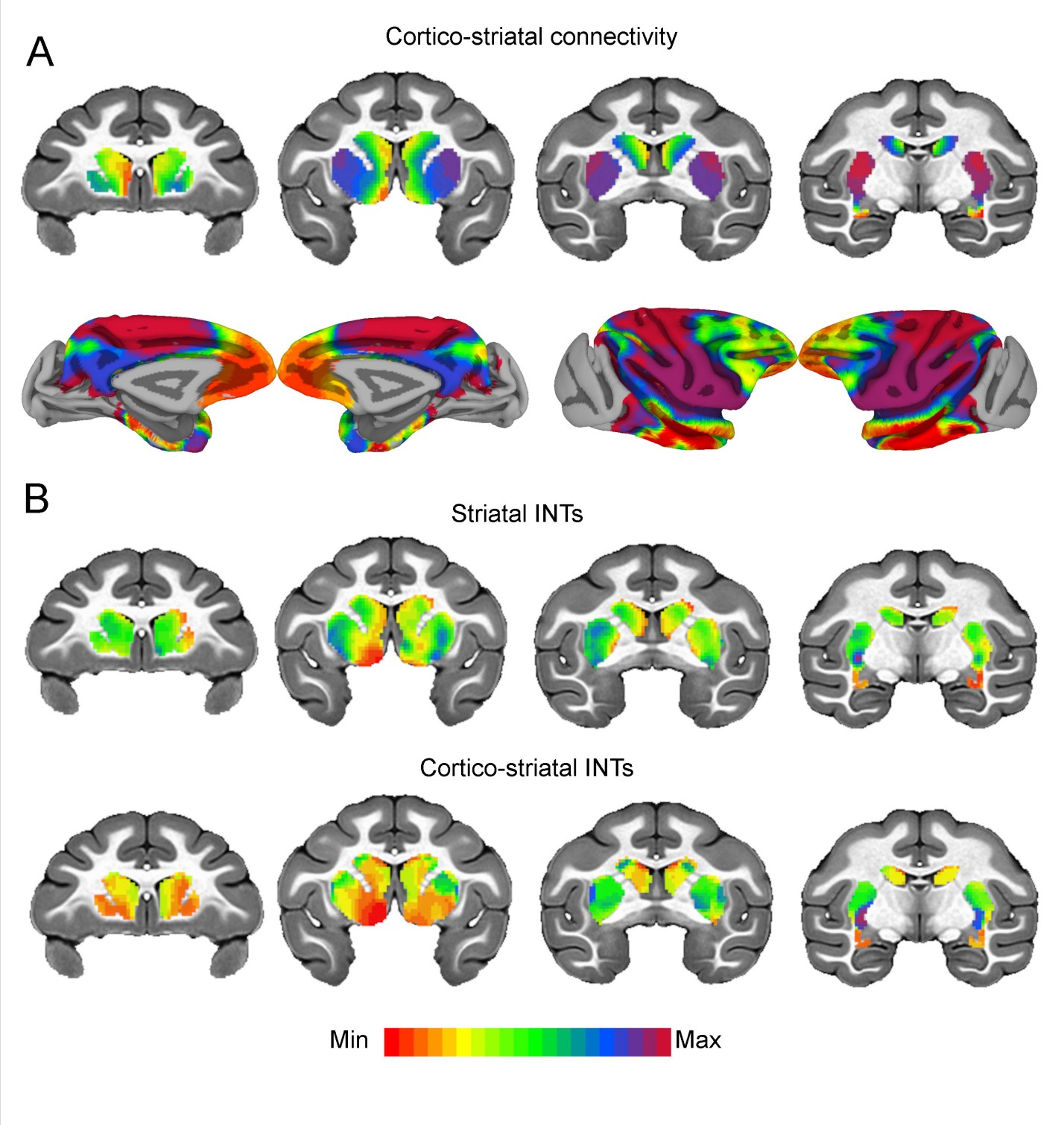

**Figure 5.** Cortico-striatal functional connectivity (FC) is related to intrinsic neural timescales (INTs). Group-averaged INT maps and FC gradients (Left) were estimated at the group level (N=9). Gradient 1 (G1) and Gradient 2 (G2) were estimated using a cosine similarity affinity computation of the cortico-striatal FC followed by diffusion mapping (Note: G1 and G2 are the first components that describe the axes of largest variance). Only the gradient with the highest correlation to the striatal INT map was plotted. (**A**) The striatal FC gradient estimated from cortico-striatal connectivity (Top) and its projection onto the cortex (Bottom) (Note: the same value range was used). The FC patterns underlying the cortico-striatal gradient matched the anatomical connectivity derived from animal invasive tracing. (**B**) Striatal INTs (Top) and cortical INTs projection based on cortico-striatal FC (Bottom). The striatal and cortico-striatal INT maps were related (r=0.45). Striatal INTs were related to G1 estimated from cortico-striatal connectivity (r=0.38). The

*Figure 5 continued on next page*

*Figure 5 continued*

color bar indicates the position along the FC gradient (values are on an arbitrary scale) or INT values (Note: since the hemodynamic INTs were estimated as the sum of ACF values in the initial positive period, this measurement considers both the number of lags and the magnitude of the ACF values. As a result, the INT does not have a time unit and it is used as a proxy of the timescale of that area, with larger values reflecting longer timescales). The value range differs across panels unless otherwise stated.

The online version of this article includes the following figure supplement(s) for figure 5:

**Figure supplement 1.** Intrinsic neural timescales (INTs) and functional connectivity (FC) gradients are monotonically related in the striatum.

is, a spatially localized increase in the fMRI signal directly and monotonically reflects an increase in neural activity, with the mean onset time of the BOLD signal relative to the neural activation being 2 s (*Logothetis et al., 2001*). As a result of sensory stimulation, BOLD signal is delayed by ~2–3 s, followed by a ramp of 6–12 s to a plateau or peak value. Local field potentials (LFPs), and to a lesser extent multiunit activity, can predict the BOLD signal when convolved with the neuro-vascular impulse response function. In sum, the hemodynamic response is roughly a low-pass-filtered expression of the total neural activity; with LFPs, however, having a greater contribution to the signal. This has several implications for the interpretation of hemodynamic timescales: (1) they are more likely to reflect timescales related to LFPs rather than spiking activity; (2) the time constants estimated from spiking activity might not be linearly related to those estimated from fMRI. A further complication for the interpretability of hemodynamic timescales is that INTs have only been estimated from spiking activity, but not LFPs. The exact relationship between timescales derived from spiking activity, LFPs and the BOLD signal remains an empirical question. We hypothesize that hemodynamic timescales generally increase monotonically with those estimated from electrophysiology. In that respect, *Watanabe et al., 2019* bring preliminary evidence—that is, timescales derived from electroencephalography (EEG) and fMRI have been shown to differ by 2 orders of magnitude. This is reasonable considering the BOLD time course—when convolved with the hemodynamic response function, the INTs estimated from EEG were similar in magnitude to those estimated from fMRI. Nevertheless, we show that temporal hierarchies are a general principle of organization, irrespective of the method used to estimate them.

To further demonstrate the potential of INTs estimated from fMRI, we extend the current literature by providing a finer description of the hierarchy of visual INTs compared to those previously described with electrophysiology (*Murray et al., 2014*). Remarkably, areas along the dorsal visual pathway follow the INT hierarchical ordering predicted by functional specialization and connectivity (*Kravitz et al., 2011*), with primary visual areas showing the fastest timescales and prefrontal areas showing the longest. Our findings bring further evidence that the dorsal pathway is not a unitary system, but rather subdivided into multiple parallel pathways (*Binkofski and Buxbaum, 2013*; *Kravitz et al., 2011*). This demonstrates the importance of fine-grained, large-scale descriptions of the brain for uncovering multiple hierarchies.

We investigated timescales and connectivity topographies at different levels of description to probe whether one aspect of the multidimensional functional architecture of the brain is related to temporal structure. We show that INT topographies are closely related to the overlapping FC gradient dimensions. Previous studies have demonstrated that FC gradient dimensions in sensory cortex corresponded with known organizational principles such as eccentricity and polar angle (*Haak et al., 2018*). Moreover, FC gradients followed M1's somatotopic map. Our findings put forward a significantly more unified framework that generalizes over the entire brain. INTs form a topographic organizational principle ranging from sensory to associative cortex. While other organizational principles can and likely always will be represented across topographic networks of the brain, intrinsic timescales appear to be special in their general extent.

It is noteworthy that there are crucial methodological differences between the gradient and INT estimation approaches: (1) the gradient estimation pipeline consists of multiple steps to reduce the dimensionality of the data, and consequently, the amount of noise (e.g., the FC matrix was made sparse; applying the cosine similarity kernel to build an affinity matrix; and applying a nonlinear dimensionality reduction technique to estimate the gradients); (2) the gradients were estimated bilaterally. In comparison to gradients, where the spatial correlation between the voxels' time series is just the initial step in the pipeline, the INT measure consists of the temporal autocorrelation of individual voxels. No extra steps have been taken for noise reduction. Hence, while gradients are the result of a complex processing pipeline, INTs are a relatively simple calculation. As a result, there is a higher amount of

variability in the INT maps which could potentially lead to the underestimation of the FC-INT relationship. With respect to the second point, the symmetry of the gradients could be a byproduct of estimating the gradients bilaterally. Other studies e.g., *Haak et al., 2018*; *Przezdzik et al., 2019* have derived gradients for each hemisphere individually and no major asymmetries have been reported. We estimated the gradients bilaterally because of an issue that is inherent with experimental ultrahigh fields. Specifically, RF transmit field (or B1) inhomogeneities lead to asymmetric signal intensities. We were worried that these effects might introduce asymmetries between the two hemispheres. We have reason to believe that this issue is partially being taken care of by estimating the gradients bilaterally, but the INTs are still susceptible to these effects. This could also result in the underestimation of the FC-INT relationship since there are visible asymmetries in the INT maps.

One potential limitation of the current study is using anesthetized subjects. Although obtaining high-resolution images of awake NHPs at ultrahigh fields would be ideal, it is still an enormous technical challenge that the field is working to overcome. Using our low anesthesia strategy, we were able to obtain a relatively large sample size of NHPs (because extensive training was not required). It is important to note that it was previously shown that our anesthesia protocol demonstrates no observable biases to whole-brain connectivity analysis (*Yacoub et al., 2020*). Moreover, it has been shown that this protocol has a relatively low impact on reproducibility (*Autio et al., 2021*). To our knowledge, the largest effect of vasodilative anesthetics (such as isoflurane) is an overall blunting of responses which is counteracted by ultrahigh field MRI. Finally, besides the possible effects on the hemodynamic response, anesthesia introduces a crucial difference between the current study and previous research—while in previous studies, it was not controlled for the subjects engaging in various cognitive processes, in anesthetized subjects, this confound is potentially removed. Overall, future research should replicate current results to exclude a possible effect of anesthesia.

We show that resting-state fMRI in NHPs is crucial for bridging the gap between human and animal models, but also across the various techniques of investigating the brain. Nevertheless, macaques have a disadvantage as a model organism for fMRI because of their small head size (~6% of the human brain) which makes the use of standard field strength magnets problematic for acquiring high-resolution data. As a result, collecting high-resolution, high SNR whole-brain data has been extremely difficult (*Yacoub et al., 2020*). We show here that one way of overcoming this problem is to use ultrahigh field strengths in combination with high channel count receiver arrays (*Lagore et al., 2021*; *Uğurbil et al., 2019*; *Yacoub et al., 2020*). Ultrahigh magnetic fields not only enhance SNR but also increase the BOLD contrast, especially for the microvascular BOLD contributions—which leads to mapping signals with higher precision to sites of neuronal activity (*Ugurbil, 2016*; *Uğurbil, 2018*). In conclusion, our unique approach made it possible to obtain whole-brain high-resolution data to investigate FC and INTs across both cortical and subcortical structures.

INTs, like AC and FC, are one way of characterizing the organization of the brain. Critically, combined maps of all three will require a nonhuman animal model such as the macaque, as AC cannot be directly measured in humans (*Grier et al., 2020*). It is clear that AC, FC, and INTs are related, although the precise nature of the relationships is somewhat speculative. For instance, vertex FC strength is related to human INTs estimated from fMRI (*Raut et al., 2020*). Our own results show that cortico-striatal INTs and FC gradients broadly track AC. AC establishes the pathways which the signals underpinning FC use. However, the fact that the pathways exist and are used does not tell us what types of information transformations occur along the way. Through cascaded processing, longer and longer TRWs can be built, measured as longer INTs. Thus, we expect that the space of potential INTs is constrained by both AC and FC. Future exploration of these organizational principles, including using pharmacological interventions and electrical stimulation, will likely clarify the causal relationships at play.

We show that INTs are topographically organized along FC gradients throughout the NHP brain. Although we focused our analysis on the frontal and parietal lobes, and the striatum, we also showed that a whole-brain analysis recapitulates the same conclusion: resting-state temporal and spatial dynamics are closely related. Future investigations should examine this relationship in other cortical and subcortical brain areas. A promising avenue would be to examine the FC-INT link throughout the brain across different levels of parcellation (and different types of parcellations). While we show that intrinsic timescales are related to FC, other studies have related them to other anatomical and functional markers (e.g., see *Ito et al., 2020*; *Watanabe et al., 2019*). Hence, further investigations and careful experimentation are needed to understand (1) how temporal dynamics relate to other

functional and anatomical characteristics across the brain; (2) their functional relevance and translational potential.

## Materials and methods

### Animal preparation

We obtained data from nine adult macaque monkeys (*Macaca fascicularis*: five female, and *Macaca mulatta*, three male and one female). Weights ranged from 3.0 kg to 9.4 kg. Experimental procedures were carried out in accordance with the University of Minnesota Institutional Animal Care and Use Committee and the National Institute of Health standards for the care and use of NHPs. All subjects were fed ad libitum and pair-housed within a light and temperature-controlled colony room. Animals were not water restricted. None of the subjects had any prior implant or cranial surgery. Animals were fasted for 14–16 hr prior to imaging.

On scanning days, anesthesia was first induced by intramuscular injection of atropine (0.5 mg/kg), ketamine hydrochloride (7.5 mg/kg), and dexmedetomidine (13 µg/kg). Subjects were transported to the scanner anteroom and intubated using an endotracheal tube. Initial anesthesia was maintained using 1.0%–2% isoflurane mixed with oxygen (1 L/m during intubation and 2 L/m during scanning to compensate for the 12-m length of the tubing used). For functional imaging, the isoflurane level was lowered to 1%.

Subjects were placed onto a custom-built coil bed with integrated head fixation by placing stereotactic ear bars into the ear canals. The position of the animal corresponds to the sphinx position. Experiments were performed with animals freely breathing. An initial bolus injection of 1.5 µg/kg fentanyl was administered IV followed by a continuous administration of 3 µg/kg/hr using a syringe pump. Rectal temperature (~99.6F), respiration (10–15 breaths/min), end-tidal $CO_2$ (25–40), electrocardiogram (70–150 bpm), and $SpO_2$ (>90%) were monitored using an MRI compatible monitor (IRAD-IMED 3880 MRI Monitor, USA). Temperature was maintained using a circulating water bath as well as chemical heating pads and padding for thermal insulation.

### Data acquisition

All data were acquired on a passively shielded 10.5 T, 88-cm diameter clear bore magnet coupled to Siemens gradients ('SC72' body gradients operating at a slew rate of 200 mT/m/s, and 70 mT/m maximal strength), and electronics (Magnetom 10.5 T Plus) (Siemens, Erlangen, Germany). Within the gradient set and the bore-liner, the space available for subject insertion had a 60-cm diameter.

The 10.5-T system operates on the E-line (E12U) platform which is directly comparable to clinical platforms (3 T Prisma/Skyra, 7 T Terra). As such, the user interface and pulse sequences were identical to those running on clinical platforms. A custom in-house built and designed RF coil with an 8-channel transmit/receive end-loaded dipole array of 18-cm length (individually) combined with a close-fitting 16-channel loop receive array head cap, and an 8-channel loop receive array of 50×100 mm² size located under the chin (*Lagore et al., 2021*). The size of 14 individual receive loops of the head cap was 37 mm with two larger ear loops of 80 mm—all receiver loops were arranged in an overlapping configuration for nearest neighbor decoupling. The resulting combined 32 receive channels were used for all experiments and supported threefold acceleration in phase encoding direction. The coil holder was designed to be a semi-stereotaxic instrument holding the head of the animal in a centered sphinx position via customized ear bars. The receive elements were modeled to adhere as close to the surface of the animal's skulls as possible. Transmit phases for the individual transmit channels were fine-tuned for excitation uniformity for one representative mid-sized animal and the calculated phases were then used for all subsequent acquisitions. Magnetic field homogenization (B0 shimming) was performed using a customized field of view with the Siemens internal 3D mapping routines. Multiple iterations of the shims (using the adjusted FOV shim parameters) were performed, and further fine adjustment was performed manually on each animal. Third-order shim elements were ignored for these procedures.

In all animals, a B1+ (transmit B1) fieldmap was acquired using a vendor provided flip angle mapping sequence and then power calibrated for each individual. Following B1+ transmit calibration, 3–5 averages (23 min) of a T1 weighted magnetization prepared rapid acquisition gradient echo protocol (3D MP-RAGE) were acquired for anatomical processing (TR=3300 ms, TE=3.56 ms, TI=1140

ms, flip angle=5°, slices=256, matrix=320×260, acquisition voxel size=0.5×0.5×0.5 mm³). Images were acquired using in-plane acceleration GRAPPA=2. A resolution and FOV matched T2 weighted 3D turbo spin-echo sequence (variable flip angle) was run to facilitate B1 inhomogeneity correction.

Before the start of the functional data acquisition, five images were acquired in both phase-encoding directions (R/L and L/R) for offline EPI distortion correction. Five to six runs of fMRI time series, each consisting of 700 continuous 2D multiband (MB)/Simultaneous Multi SLice (SMS) EPI (*Moeller et al., 2010*; *Setsompop et al., 2012*; *Uğurbil et al., 2013*) functional volumes (TR=1110 ms; TE=17.6 ms; flip angle=60°, slices=58, matrix=108×154; FOV=81×115.5 mm²; acquisition voxel size=0.75×0.75×0.75 mm³) were acquired. Images were acquired with a left-right phase encoding direction using in plane acceleration factor GRAPPA=3, partial Fourier=7/8th, and MB or simultaneous multislice factor=2 (i.e., the number of simultaneously excited slices=2). Since macaques were scanned in sphinx positions, the orientations noted here are what is consistent with a (head first supine) typical human brain study (in terms of gradients) but translate differently to the actual macaque orientation.

## Image preprocessing

Image processing was performed using a custom pipeline relying on FSL (*Jenkinson et al., 2012*), ANTs (*Avants et al., 2014*; *Avants et al., 2011*), AFNI (*Cox, 1996*), and a heavily modified CONN (*Whitfield-Gabrieli and Nieto-Castanon, 2012*) toolbox. Images were first motion corrected using *mcflirt* (registration to the first image). Motion parameters never exceeded 0.5 mm or 0.4° rotation except in one run of one animal, which was discarded (the animal experienced anesthesia-induced emesis during the last run). Images were slice-time corrected and EPI distortion corrected using *topup*. High magnetic fields and large matrix sizes are commonly associated with severe EPI distortions. Anatomical images were nonlinearly warped into the 'National Institute of Mental Health Macaque Template' (NMT) (*Seidlitz et al., 2018*) template using ANTs and 3DQwarp in AFNI. The distortion correction, motion correction, and normalization were performed using a single sinc interpolation. Images were spatially smoothed (FWHM=2 mm), linear detrended, denoised using a linear regression approach including heart rate and respiration, as well as a five-component nuisance regressor of the masked white matter and cerebrospinal fluid and band-pass filtering (0.008–0.09 Hz) (*Hallquist et al., 2013*).

## Intrinsic neural timescales

Based on previous studies (*Murray et al., 2014*; *Watanabe et al., 2019*), INTs were calculated by estimating the magnitude of the autocorrelation of the preprocessed functional data. At the single-subject level, the INT of each voxel was quantified as follows. For each run, the ACF of each voxel's fMRI signal was estimated. The ACF is the correlation of $y_t$ (time series) and $y_{t+k}$ (lagged time series), where k represents the time lag in steps of TR. Next, the sum of ACF values was calculated in the initial positive period. The upper limit of this period was set at the point where the ACF first becomes negative as the time lag increases. The result was used as an index of the INT of the brain region. At the single-subject level, the number of INT maps was dependent on the number of functional runs. The maps were not smoothed, because individual variation in FC gradients and INTs has been shown to have functional significance (*Przezdzik et al., 2019*; *Watanabe et al., 2019*).

To examine the correspondence between fMRI-based and electrophysiology-based INT hierarchies, the values of individual voxels were averaged within areas. The areas were defined based on the Cortical Hierarchy Atlas of the Rhesus Macaque (CHARM) (*Jung et al., 2021*) and Subcortical Atlas of the Rhesus Macaque (SARM) (*Hartig et al., 2021*) to closely match the corresponding electrophysiological recording sites (see *Supplementary file 1*). In some cases, since the CHARM parcellation did not match the recording sites, the regions of interest were drawn.

Since the hemodynamic INTs were estimated as the sum of ACF values in the initial positive period, this measurement considers both the number of lags and the magnitude of the ACF values. As a result, the hemodynamic INTs do not have a time unit and are used as a proxy of the timescale of that area, with higher values reflecting longer timescales. We have additionally provided the estimated time constants associated with the INT hierarchies that have been previously described with electrophysiological recordings by multiplying the 0-crossing lag by the TR (see *Figure 1—figure supplement 2* for an estimate of the time constants).

## Stability and similarity

The intra-subject stability was quantified by calculating the pairwise correlation (i.e., Pearson's r) between the runs. The individual stability score was computed by averaging the Fisher's Z-transformed correlation coefficients across pairwise comparisons. For every subject, the maps were averaged across runs to quantify the inter-subject similarity. The individual similarity score was computed by averaging the subject's Fisher's Z-transformed correlation coefficients with all the other subjects.

To quantify the stability of the hierarchies, they were re-computed for groups of four subjects. The resulting ranking was correlated (i.e., Spearman's rank correlation) with the group-level ordering of the cortical/subcortical regions. To estimate the stability score, this procedure was repeated for all the possible combinations of subjects (i.e., n=126). The stability score of any hierarchy was computed by averaging the correlation coefficients across steps (i.e., n=126).

## FC gradients

We computed FC gradients using BrainSpace, a toolbox for the identification, visualization, and analysis of macroscale gradients in neuroimaging data sets (*Vos de Wael et al., 2020*).

### The input matrix

The input matrix consisted of the voxel-wise FC between or within areas. The Pearson correlation coefficient between the time-series of every pair of voxels within and/or between areas was computed. This resulted in a 2D matrix of correlation coefficients, with every voxel being represented by an n-dimensional row vector (with n reflecting the number of connections). The connectivity matrices were computed for each subject and run.

### Generating gradients

The procedure for generating FC gradients was thoroughly described in *Vos de Wael et al., 2020*. The input matrix was made sparse (10% sparsity) and the cosine similarity kernel was applied to build the affinity matrix. Next, diffusion mapping was applied using the BrainSpace basic parameters. The resulting gradients were plotted on the cortical surface of the NMT template (*Seidlitz et al., 2018*) using AFNI (*Cox, 1996*) and SUMA (*Saad and Reynolds, 2012*; *Saad et al., 2004*). The gradients were generated bilaterally. We only report the gradient most correlated with the INT map, assuming that ensuing gradients reflect other principles of functional organization.

### Cortical projection

To probe the relationship between the striatal FC gradients and the cortex, the striatal FC gradients were projected onto the cortex according to the highest-correlated voxels (*Haak et al., 2018*). To reduce noise and improve interpretability, the resulting maps were smoothed using AFNI (*Cox, 1996*).

## Projecting INTs based on FC

To examine the relationship between INTs and FC between areas (e.g., between cortex and striatum), cortical INTs were projected onto a striatal map based on cortico-striatal FC. For every striatal voxel, the top 10 highest functionally correlated cortical voxels were identified. Next, the INTs associated with these cortical voxels were extracted. Finally, the average associated cortical INT value was computed for each striatal voxel and represented as a map.

## Acknowledgements

The authors thank Steve Jungst, Gregor Adriany, and Russell Lagore for continuing support with our coils and hardware. The authors thank Jen Holmberg, Brenna Knaebe, and Mrunal Zambre, for support with animal care and data acquisition. This work was supported by NIH grants P41 EB027061 (to JZ and KU), R01 MH118257 (to SRH), R56 EB031765 (to JZ), R01 MH128177 (to JZ), from the Digital Technologies Initiative (to JZ), from the Minnesota Institute of Robotics (to JZ), two Young Investigator Awards from the Brain & Behavior Research Foundation to SRH and AZ, a P30DA048742 (to JZ, SRH, and AZ), and a UMN AIRP award to JZ, SRH, and AZ.

## Additional information

### Funding

| Funder | Grant reference number | Author |
|---|---|---|
| National Institutes of Health | P41 EB027061 | Kamil Ugurbil<br>Jan Zimmermann |
| National Institutes of Health | R01 MH118257 | Sarah R Heilbronner |
| National Institutes of Health | R56 EB031765 and R01 EB031765 | Jan Zimmermann |
| National Institutes of Health | R01 MH128177 | Jan Zimmermann |
| Digital Technologies Initiative | | Jan Zimmermann |
| Minnesota Institute of Robotics | | Jan Zimmermann |
| Brain & Behavior Research Foundation | Young Investigator Awards | Anna Zilverstand<br>Sarah Heilbronner |
| National Institutes of Health | P30DA048742 | Anna Zilverstand<br>Sarah Heilbronner<br>Jan Zimmermann |
| UMN AIRP award | | Anna Zilverstand<br>Sarah Heilbronner<br>Jan Zimmermann |

The funders had no role in study design, data collection and interpretation, or the decision to submit the work for publication.

### Author contributions

Ana MG Manea, Conceptualization, Formal analysis, Investigation, Methodology, Software, Visualization, Writing - original draft, Writing – review and editing; Anna Zilverstand, Conceptualization, Funding acquisition, Writing – review and editing; Kamil Ugurbil, Funding acquisition, Writing – review and editing; Sarah R Heilbronner, Conceptualization, Funding acquisition, Investigation, Resources, Writing – review and editing; Jan Zimmermann, Conceptualization, Data curation, Formal analysis, Funding acquisition, Investigation, Methodology, Resources, Software, Supervision, Writing - original draft, Writing – review and editing

### Author ORCIDs

Ana MG Manea (iD) http://orcid.org/0000-0002-4786-9657
Anna Zilverstand (iD) http://orcid.org/0000-0002-4889-9700

### Ethics

Experimental procedures were carried out in accordance with the University of Minnesota Institutional Animal Care and Use Committee and the National Institute of Health standards for the care and use of nonhuman primates. Protocol IDs: 2005-38127A 2005-38135A 1911-37623A.

### Decision letter and Author response

Decision letter https://doi.org/10.7554/eLife.75540.sa1
Author response https://doi.org/10.7554/eLife.75540.sa2

## Additional files

### Supplementary files

• Supplementary file 1. The areas were defined based on a cortical and subcortical atlas. The areas in *Figure 1* were defined based on the Cortical Hierarchy Atlas of the Rhesus Macaque (CHARM)

(*Jung et al., 2021*) and Subcortical Atlas of the Rhesus Macaque (SARM) (*Hartig et al., 2021*) to closely match the corresponding electrophysiological recording sites. For some of the cortical regions of interest, the CHARM parcellation did not match the recording sites. As a result, the areas were defined according to the respective descriptions (Note: this is the case whenever "Custom" is indicated in the table). Abbreviations: S1/S2 (primary/secondary somatosensory cortex), MT (middle temporal area), LIP (lateral intraparietal area), OFC (orbitofrontal cortex), LPFC (lateral prefrontal cortex), ACC (anterior cingulate cortex), PMd (dorsal premotor cortex), LOFC (lateral OFC), DLPFC (dorso-lateral PFC), VLPFC (ventro-lateral PFC), PFp (polar prefrontal cortex), vmPFC (ventro-medial PFC), sgACC (subgenual ACC), pgACC (pregenual ACC), dACC (dorsal ACC), GPe (external globus pallidus), STN (subthalamic nucleus).

• Supplementary file 2. Areas defined based on a cortical atlas. The areas in *Figure 2* were defined based on the Cortical Hierarchy Atlas of the Rhesus Macaque (CHARM) (*Jung et al., 2021*) to match the hierarchies described by *Kravitz et al., 2011*. Abbreviations: DLPFC (dorso-lateral prefrontal cortex), VIP (ventral intraparietal area), MIP (medial intraparietal area), rIPL (rostral inferior parietal lobule), MT (middle temporal area), MST (medial superior temporal area), LIP (lateral intraparietal area), A46 (area 46), A8A (area 8 A), PMd (dorsal premotor cortex), PMv (ventral premotor cortex).

• Transparent reporting form

## Data availability

The functional connectivity gradient maps and the timescale maps have been uploaded to figshare. Functional connectivity gradients: https://doi.org/10.6084/m9.figshare.19189331 Intrinsic neural timescales: https://doi.org/10.6084/m9.figshare.19197026.

The following dataset was generated:

| Author(s) | Year | Dataset title | Dataset URL | Database and Identifier |
|---|---|---|---|---|
| Manea AMG, Zilverstand A, Uğurbil K, Heilbronner SR, Zimmermann J | 2021 | Resting-state functional connectivity gradients and intrinsic neural timescales | https://doi.org/10.6084/m9.figshare.19189331 | figshare, 10.6084/m9.figshare.19189331 |

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
