## [Editor Report]

Neural activity measured in both electrophysiological and functional neuroimaging experiments are often temporally correlated, and the timescales of such correlation in ongoing neural activity, or intrinsic neural timescales, show a hierarchical pattern across the cortical surface. The present study establishes a close link between these timescales and functional connectivity in the brains of non-human primates, suggesting that temporal autocorrelation is an important organizing feature of large-scale neural activity.

---

## [Decision Letter]

**Decision letter after peer review:**

Thank you for submitting your article "Intrinsic timescales as an organizational principle of neural processing across the whole rhesus macaque brain" for consideration by *eLife*. Your article has been reviewed by 3 peer reviewers, including Daeyeol Lee as Reviewing Editor and Reviewer #1, and the evaluation has been overseen by Timothy Behrens as the Senior Editor.

Essential revisions:

1. Effects of anesthesia. Compared to previous human neuroimaging studies and NHP electrophysiological experiments, a significant weakness of this study is that the experiments were performed in anesthetized animals. This is even more problematic, since the analyses in this study mostly focus on the fronto-parietal network, which might be more affected by anesthetics. Although it has been shown that the FC can be studied in anesthetized animals, how specific methods of anesthesia might influence by the measures of INT should be discussed more thoroughly.

2. INT measure. They authors put forth the INT measure as related to intrinsic timescale. INT is defined as the integrated area under the autocorrelation function (ACF) up until the first point where the ACF goes below zero. This is different than how intrinsic timescale has been measured in single-neuron spike trains or in prior fMRI studies. While a longer timescale would be expected to increase INT, the problem is that INT (as an integrated area) combines effects of autocorrelation timescale and autocorrelation amplitude.

– It would be insightful to visualize INT properties at the whole-brain or whole-cortex level (instead of only a single lobe), including (i) INT values themselves, (ii) the lag-one autcorrelation value reflecting autocorrelation amplitude, and (iii) the zero-crossing lag time used to compute INT.

– A highly relevant paper (which is not currently cited) is Ito et al., (2020) NeuroImage, "A cortical hierarchy of localized and distributed processes revealed via dissociation of task activations, connectivity changes, and intrinsic timescales". Figure 5 of Ito shows a cortex-wide map of intrinsic timescale as defined in single-neuron studies (i.e. fitting time constant of decay). Figure 6 then shows this is related to cortical hierarchy as reflected in the T1w/T2w map (which in principle could be tested by the authors here too). Ito's analysis was performed on the parcellated timeseries, not the voxel level as in the present study, which is a notable methodological difference.

– That INT combines effects of timescale and amplitude would not be a problem if the autocorrelation amplitude does not vary across brain regions. However, it appears that it does for whatever reasons (neural and/or in fMRI measurement such as SNR). A relevant preprint is by Shinn et al., (2021) bioRxiv, “Spatial and temporal autocorrelation weave human brain networks”. In human cortex, again using parcellated timeseries, Figure 1F there shows systematic variation across cortical parcels of the lag-1 autocorrelation value. In the present study, it is currently unknown whether INT is reflecting regional differences in autocorrelation timescale (as interpreted), amplitude (not considered), or both.

– Contribution of autocorrelation amplitude to INT may potentially explain why a cortex-wide map of INT does not follow an expected hierarchy as much the more restricted views within one lobe as the current manuscript focuses. For instance, Figure 1 shows that INT values for somatosensory cortex (Figure 1A) are larger than association regions (Figure 1B). Is this potentially due to autocorrelation amplitude being larger in somatosensory cortex?

– Perhaps some smoothing or parcellation would be required to better tease apart autocorrelation timescale from autocorrelation amplitude.

– A highly relevant paper (which is not currently cited) is Ito et al., (2020) NeuroImage, "A cortical hierarchy of localized and distributed processes revealed via dissociation of task activations, connectivity changes, and intrinsic timescales". Figure 5 of Ito shows a cortex-wide map of intrinsic timescale as defined in single-neuron studies (i.e. fitting time constant of decay). Figure 6 then shows this is related to cortical hierarchy as reflected in the T1w/T2w map (which in principle could be tested by the authors here too). Ito's analysis was performed on the parcellated timeseries, not the voxel level as in the present study, which is a notable methodological difference.

– That INT combines effects of timescale and amplitude would not be a problem if the autocorrelation amplitude does not vary across brain regions. However, it appears that it does for whatever reasons (neural and/or in fMRI measurement such as SNR). A relevant preprint is by Shinn et al., (2021) bioRxiv, "Spatial and temporal autocorrelation weave human brain networks". In human cortex, again using parcellated timeseries, Figure 1F there shows systematic variation across cortical parcels of the lag-1 autocorrelation value. In the present study, it is currently unknown whether INT is reflecting regional differences in autocorrelation timescale (as interpreted), amplitude (not considered), or both.

– Contribution of autocorrelation amplitude to INT may potentially explain why a cortex-wide map of INT does not follow an expected hierarchy as much the more restricted views within one lobe as the current manuscript focuses. For instance, Figure 1 shows that INT values for somatosensory cortex (Figure 1A) are larger than association regions (Figure 1B). Is this potentially due to autocorrelation amplitude being larger in somatosensory cortex?

– Perhaps some smoothing or parcellation would be required to better tease apart autocorrelation timescale from autocorrelation amplitude.

3. Does the calculation of fMRI-based neuronal time constants obscure the unit of time? True comparison with ephys data is not possible without clarifying the relationship of the two quantities compared. In the ephys measurements time constants are in units of seconds and often below 1s. In contrast, BOLD response has a sluggish time course (tens of seconds) due to the properties of the hemodynamic response function. The smoothing of spiking and field-potential activity with the HRF introduces substantial auto-correlation in BOLD and is expected to reduce our ability to distinguish small differences of time constants discovered with ephys. Because the analyses in this paper do not explore the complications caused by the slow and noisy BOLD measurements, it is impossible to know if the observed temporal hierarchy has the same nature and origin as those reported with ephys. It would be useful to perform additional analyses and modeling that clarifies this missing link. If that is not possible, at the very least I would recommend explicit reporting of the units of time constants based on BOLD in the figures, and discussing if the differences of BOLD time constants across regions match the differences of spiking activity time constants in previous publications. Also, throughout the paper, figures should clearly indicate the unit for INT.

4. Functional connectivity gradients: Figures 3 and 4 rely on functional connectivity gradients calculated within a single lobe, against which INT topography is correlated. On such a restricted geometry as a single lobe, a functional connectivity gradient may be reflecting a simpler property, namely the geometry of the restricted cortical sheet. In other words, given the sheet geometry of the frontal lobe, does an anterior-posterior topography fall out naturally as the first gradient (e.g. with distance-dependent falloff) and medial-lateral as second gradient? If so, it is difficult to strongly interpret these results as linking INT to functional connectivity when the gradient is a generic consequence of the sheet geometry. In human neuroimaging such functional gradients are typically calculated at the whole-cortex level which reveals less trivial topographies (e.g. Margulies et al., 2016, PNAS). These results and interpretations should be considered in light of this concern.

5. There is no strong relationship between within area FC and INT maps in figure 4. Unlike the FC maps that seem to monotonically change along a cortical axis, the INT maps seem to have non-monotonic changes. For example, the INT map for OFC has maximum values both in the anterior-medial and caudal-lateral regions. Similar discrepancies between the INT and FC maps are apparent for PFC. Also, unlike the FC maps, the INT maps seem asymmetrical in the two hemispheres. Unfortunately, the visualization and stats are too dense to probe what is going on. Deeper analyses and explanation are necessary (please see the next comment for a potential solution).

6. Many fine level discrepancies are also observed between striatal INTs and cortico-striatal INTs. In the current figures, it is impossible for readers to know if the variations and discrepancies are within the range expected from noise in the data. The alternative is that some of the variations are meaningful but ignored by the authors to keep the narrative simple. Either way, the authors are encouraged to improve their presentation method and analyses in order to provide adequate information for readers. One possibility is to use old-fashioned scatter plots with error bars, where individual data points are different locations within the striatum and the axes are striatal INTs and cortico-striatal INTs. The authors can choose how they'd like to divide striatum into distinct sub-regions (e.g., a simple 3D grid or functional divisions). Similar plots would be useful for figure 4, and more generally anywhere that the authors report a correlation between INTs and FCs. 7.

7. The authors should provide more detailed information about how the hierarchy shown in the x-axis of Figure 1 was computed, so that this can be replicated by other investigators. Some schematic diagram for cortical hierarchy might be also helpful.

8. The paper would have more value if the same analyses were carried out in other brain areas, such as the temporal cortex, that are not currently covered. If such results are not included, the authors should at least provide a reason why the paper focuses only on the frontal and parietal regions.

---

## [Author Response]

Essential revisions:1. Effects of anesthesia. Compared to previous human neuroimaging studies and NHP electrophysiological experiments, a significant weakness of this study is that the experiments were performed in anesthetized animals. This is even more problematic, since the analyses in this study mostly focus on the fronto-parietal network, which might be more affected by anesthetics. Although it has been shown that the FC can be studied in anesthetized animals, how specific methods of anesthesia might influence by the measures of INT should be discussed more thoroughly.

The reviewers raise an important question and concern, and we would like to apologize for this omission. In a previous paper from our lab (Yacoub et al., 2020) we were able to demonstrate that our low isoflurane + dexamethasone (Autio et al., 2021)ng autocorrelation amplitude, and (iii) the zero-crossing la anesthesia demonstrates no observable biases to whole brain connectivity analysis, and stimulation experiments in our lab (unpublished) demonstrate non altered hemodynamic response function properties. In fact, hemodynamic properties seem to be significantly more affected by the common use of contrast agents such as ferromolecules (Pelekanos et al., 2020). In our experience, the largest effect of vasodilative anesthetics (such as isoflurane) is an overall blunting of responses which the use of ultrahigh magnetic fields counteracts. We have added the following paragraph to the discussion of the manuscript for transparency:

“One potential limitation of the current study is using anesthetized subjects. Although obtaining high resolution images of awake NHPs at ultrahigh fields would be ideal, it is still an enormous technical challenge that the field is working to overcome. Using our low anesthesia strategy, we were able to obtain a relatively large sample size of NHPs (because extensive training was not required). It is important to note that it was previously shown that our anesthesia protocol demonstrates no observable biases to whole-brain connectivity analysis (Yacoub et al., 2020). Moreover, it has been shown that this protocol has a relatively low impact on reproducibility (Autio et al., 2021). To our knowledge, the largest effect of vasodilative anesthetics (such as isoflurane) is an overall blunting of responses which is counteracted by ultrahigh field MRI. Finally, besides the possible effects on the hemodynamic response, anesthesia introduces a crucial difference between the current study and previous research – while in previous studies it was not controlled for the subjects engaging in various cognitive processes, in anesthetized subjects, this confound is potentially removed. Overall, future research should replicate current results to exclude a possible effect of anesthesia.”

2. INT measure. They authors put forth the INT measure as related to intrinsic timescale. INT is defined as the integrated area under the autocorrelation function (ACF) up until the first point where the ACF goes below zero. This is different than how intrinsic timescale has been measured in single-neuron spike trains or in prior fMRI studies. While a longer timescale would be expected to increase INT, the problem is that INT (as an integrated area) combines effects of autocorrelation timescale and autocorrelation amplitude.– It would be insightful to visualize INT properties at the whole-brain or whole-cortex level (instead of only a single lobe), including (i) INT values themselves, (ii) the lag-one autcorrelation value reflecting autocorrelation amplitude, and (iii) the zero-crossing lag time used to compute INT.– A highly relevant paper (which is not currently cited) is Ito et al., (2020) NeuroImage, "A cortical hierarchy of localized and distributed processes revealed via dissociation of task activations, connectivity changes, and intrinsic timescales". Figure 5 of Ito shows a cortex-wide map of intrinsic timescale as defined in single-neuron studies (i.e. fitting time constant of decay). Figure 6 then shows this is related to cortical hierarchy as reflected in the T1w/T2w map (which in principle could be tested by the authors here too). Ito's analysis was performed on the parcellated timeseries, not the voxel level as in the present study, which is a notable methodological difference.– That INT combines effects of timescale and amplitude would not be a problem if the autocorrelation amplitude does not vary across brain regions. However, it appears that it does for whatever reasons (neural and/or in fMRI measurement such as SNR). A relevant preprint is by Shinn et al., (2021) bioRxiv, "Spatial and temporal autocorrelation weave human brain networks". In human cortex, again using parcellated timeseries, Figure 1F there shows systematic variation across cortical parcels of the lag-1 autocorrelation value. In the present study, it is currently unknown whether INT is reflecting regional differences in autocorrelation timescale (as interpreted), amplitude (not considered), or both.– Contribution of autocorrelation amplitude to INT may potentially explain why a cortex-wide map of INT does not follow an expected hierarchy as much the more restricted views within one lobe as the current manuscript focuses. For instance, Figure 1 shows that INT values for somatosensory cortex (Figure 1A) are larger than association regions (Figure 1B). Is this potentially due to autocorrelation amplitude being larger in somatosensory cortex?– Perhaps some smoothing or parcellation would be required to better tease apart autocorrelation timescale from autocorrelation amplitude.

The possible confounding effect of a systematic regional variation of ACF amplitude is an important consideration that we have overlooked and thank the reviewers for pointing out. To further examine this issue: (1) we plotted whole-brain INTs, lag-1, and 0-crossing lag (see Figure 1 Supplement 1); (2) correlated the maps; (3) re-computed the hierarchies by using the 0crossing lag multiplied by the TR (i.e., to obtain a time estimate; see Figure 1 Supplement 2). In this study, while the amplitude as reflected by the lag-1 ACF value varies across the brain, it is not dissociated from the timescale as reflected by the 0-crossing lag. The three voxel-wise maps are highly correlated (Pearson’s r > 0.90) and re-computing the hierarchies by only using the 0-crossing lag does not change the relative “temporal” position of the various areas. Notably, the variations in the lag-1 ACF value were small, with values ranging from 0.92 to 0.95 for subcortical and 0.92 and 0.97 for cortical voxels. In consequence, it is not surprising that the hierarchies are not impacted by removing the amplitude of the ACF values from the INT measure. We thus extend the findings by Shinn et al., (2021) by showing that the lag-1 ACF value is closely related to our INT measure as well. Compared to solely using the lag-1 ACF value, our INT measure can be used to estimate the time constants of the various areas as reflected by BOLD, an added benefit in our opinion.

In contrast to previous studies, we decided to use this INT measure for two reasons: (1) it has already been validated via simultaneous EEG-fMRI recordings by Wanatabe and Masuda (2019); (2) we avoided curve fitting to reduce the adverse effects of the low sampling rate of fMRI recordings.

To clarify these important points to the reader, we added Supplement 1 and 2 to Figure 1. Moreover, we added the following paragraph to the Results section:

“Our definition of INTs combines the effects of ACF magnitude and timescale (i.e., the 0-crossing lag). Since it has previously been shown that the magnitude of the ACF values, parameterized as the lag-1 ACF value, differs across brain areas (Shinn et al., 2021), we further examined whether the observed hierarchies are driven by magnitude, timescale, or both. In line with previous results, the lag-1 ACF values systematically varied across the brain (Shinn et al., 2021). Moreover, the lag-1 ACF values and the 0-crossing lag were linearly related to INTs as operationalized in this study (see Figure 1 Supplement 1 for voxel-wise whole-brain maps of the lag-1, 0-crossing lag and INTs). The maps suggest that magnitude and timescale are closely related. To exclude the possibility that the observed hierarchies are driven by magnitude, rather than timescale, we recomputed them by using the 0-crossing lag as a proxy for INT. To obtain a time estimate of hemodynamic timescales, the 0-lag crossing lag was multiplied by the TR (i.e., 1.1s). The hierarchical ordering of both cortical and subcortical areas is consistent (see Figure 1 Supplement 2) with that based on our INT measure (see Figure 1). Hence, while the ACF magnitude is closely related to our INT index and the 0-crossing lag, temporal hierarchies are not driven by heterogeneity in the ACF magnitude.”

The reviewers bring up another interesting point about T1/T2(*) weighted anatomical images and their potential relationship to our findings. While we have acquired T2 weighted images in most of our animals they are unfortunately not purely T2 but rather a variable flip angle approach. This approach was taken due to technical difficulties porting standard sequences to our 10.5T instrument. We have recently made additional progress on allowing “purer” T2 TSE images, but for this dataset, we do not have the correct modalities available. We hope we can provide that in the future. We think there are multiple interesting parameters that should be examined in the future for their relationship with INTs, including T1/T2(*) as well as other neuroimaging and histologically derived measures.

– A highly relevant paper (which is not currently cited) is Ito et al., (2020) NeuroImage, "A cortical hierarchy of localized and distributed processes revealed via dissociation of task activations, connectivity changes, and intrinsic timescales". Figure 5 of Ito shows a cortex-wide map of intrinsic timescale as defined in single-neuron studies (i.e. fitting time constant of decay). Figure 6 then shows this is related to cortical hierarchy as reflected in the T1w/T2w map (which in principle could be tested by the authors here too). Ito's analysis was performed on the parcellated timeseries, not the voxel level as in the present study, which is a notable methodological difference.– That INT combines effects of timescale and amplitude would not be a problem if the autocorrelation amplitude does not vary across brain regions. However, it appears that it does for whatever reasons (neural and/or in fMRI measurement such as SNR). A relevant preprint is by Shinn et al., (2021) bioRxiv, "Spatial and temporal autocorrelation weave human brain networks". In human cortex, again using parcellated timeseries, Figure 1F there shows systematic variation across cortical parcels of the lag-1 autocorrelation value. In the present study, it is currently unknown whether INT is reflecting regional differences in autocorrelation timescale (as interpreted), amplitude (not considered), or both.– Contribution of autocorrelation amplitude to INT may potentially explain why a cortex-wide map of INT does not follow an expected hierarchy as much the more restricted views within one lobe as the current manuscript focuses. For instance, Figure 1 shows that INT values for somatosensory cortex (Figure 1A) are larger than association regions (Figure 1B). Is this potentially due to autocorrelation amplitude being larger in somatosensory cortex?– Perhaps some smoothing or parcellation would be required to better tease apart autocorrelation timescale from autocorrelation amplitude.3. Does the calculation of fMRI-based neuronal time constants obscure the unit of time? True comparison with ephys data is not possible without clarifying the relationship of the two quantities compared. In the ephys measurements time constants are in units of seconds and often below 1s. In contrast, BOLD response has a sluggish time course (tens of seconds) due to the properties of the hemodynamic response function. The smoothing of spiking and field-potential activity with the HRF introduces substantial auto-correlation in BOLD and is expected to reduce our ability to distinguish small differences of time constants discovered with ephys. Because the analyses in this paper do not explore the complications caused by the slow and noisy BOLD measurements, it is impossible to know if the observed temporal hierarchy has the same nature and origin as those reported with ephys. It would be useful to perform additional analyses and modeling that clarifies this missing link. If that is not possible, at the very least I would recommend explicit reporting of the units of time constants based on BOLD in the figures, and discussing if the differences of BOLD time constants across regions match the differences of spiking activity time constants in previous publications. Also, throughout the paper, figures should clearly indicate the unit for INT.

This is a very important question and hopefully future research will shed light on the exact relationship between timescales estimated from spiking activity, LFPs and BOLD. Watanabe and Masuda (2019) have shown that hemodynamic INTs differ by two orders of magnitude from those estimated from EEG. We hypothesize a similar relationship, but it is very unlikely that the hemodynamic INTs reflect only what has been previously estimated based on spiking activity. For transparency, we are adding the following paragraph to the Discussion:

“It is worth noting that the current study could not resolve the exact correspondence between the time constants estimated from spiking activity and BOLD. The relationship between neural activity and the BOLD signal has been revealed via simultaneous electrophysiological-fMRI recording-i.e., a spatially localized increase in the fMRI signal directly and monotonically reflects an increase in neural activity, with the mean onset time of the BOLD signal relative to the neural activation being 2s (Logothetis, Pauls, Augath, Trinath, and Oeltermann, 2001). As a result of sensory stimulation, BOLD signal is delayed by ~ 2-3 s, followed by a ramp of 6-12 s to a plateau or peak value. Local field potentials (LFPs), and to a lesser extent multi-unit activity (MUAs), can predict the BOLD signal when convolved with the neuro-vascular impulse response function. In sum, the hemodynamic response is roughly a low-pass-filtered expression of the total neural activity; with LFPs however, having a greater contribution to the signal. This has several implications for the interpretation of hemodynamic timescales: (1) they are more likely to reflect timescales related to LFPs rather than spiking activity; (2) the time constants estimated from spiking activity might not be linearly related to those estimated from fMRI. A further complication for the interpretability of hemodynamic timescales is that INTs have only been estimated from spiking activity, but not LFPs. The exact relationship between timescales derived from spiking activity, LFPs and the BOLD signal remains an empirical question. We hypothesize that hemodynamic timescales generally increase monotonically with those estimated from electrophysiology. In that respect, Watanabe and Masuda (2019) bring preliminary evidence – i.e., timescales derived from electroencephalography (EEG) and fMRI have been shown to differ by two orders of magnitude. This is reasonable considering the BOLD time course-when convolved with the hemodynamic response function, the INTs estimated from EEG were similar in magnitude to those estimated from fMRI. Nevertheless, we show that temporal hierarchies are a general principle of organization, irrespective of the method used to estimate them.”

Moreover, we have estimated the hemodynamic time constants by multiplying the 0-crossing lag by the TR. With those results in hand, we have re-computed the INT hierarchies from Figure 1 (see Figure 1 Supplement 2) to get an estimate for the areas that have been previously investigated with electrophysiology. With respect to the unit for INT, since the hemodynamic INTs were estimated as the sum of the ACF values in the initial positive period (our measure of INT considers both the number of lags and the magnitude of the ACF values), the INT does not have a time unit and it is used as a proxy of the timescale of that area, with higher values reflecting longer timescale. (See Figure 1 Supplement 2 for an estimate of the time constant of each area). To make this clear throughout the manuscript, we added the following paragraph to the Methods section and in the figure captions throughout the manuscript.

“Since the hemodynamic INTs were estimated as the sum of ACF values in the initial positive period, this measurement considers both the number of lags and the magnitude of the ACF values. As a result, the INT does not have a time unit and it is used as a proxy of the timescale of that area, with higher values reflecting longer timescales. We have nonetheless estimated the time constants associated with the INT hierarchies that have been previously described with electrophysiological recordings by multiplying the 0-crossing lag by the TR (see Figure 1 Supplement 2 for an estimate of the time constants).”

4. Functional connectivity gradients: Figures 3 and 4 rely on functional connectivity gradients calculated within a single lobe, against which INT topography is correlated. On such a restricted geometry as a single lobe, a functional connectivity gradient may be reflecting a simpler property, namely the geometry of the restricted cortical sheet. In other words, given the sheet geometry of the frontal lobe, does an anterior-posterior topography fall out naturally as the first gradient (e.g. with distance-dependent falloff) and medial-lateral as second gradient? If so, it is difficult to strongly interpret these results as linking INT to functional connectivity when the gradient is a generic consequence of the sheet geometry. In human neuroimaging such functional gradients are typically calculated at the whole-cortex level which reveals less trivial topographies (e.g. Margulies et al., 2016, PNAS). These results and interpretations should be considered in light of this concern.

Thank you for this insightful comment and question. Although previous studies have applied gradient approaches to the whole-brain, we avoided this approach for two reasons: (1) this is usually achieved by considerably downsampling the data, which would erase the spatial resolution provided by the ultra-high field; (2) the result would reflect the overall inter-areal connectivity differences, since inter-areal connectivity patterns are generally more pronounced than intra-areal patterns, rather than the fine-grained connectivity patterns within the individual areas. With respect to the first point, downsampling the data would potentially blur import functional details. It has been previously shown that gradients estimated for smaller regions of interest uncover important functional details across individuals – and even more, these gradients cand predict performance (Przeździk, Faber, Fernández, Beckmann, and Haak, 2019). With respect to the second point, we were not interested in capturing large-scale modes of connectivity change, but rather the functional organization of individual areas.

To address the comment raised by the reviewers, we parcellated the brain using the 6^th^ level (most fine grained) of the CHARM (Jung et al., 2021), and we calculated the FC matrix using the resulting timeseries. This resulted in a 134x134 FC matrix. This was used as input to our pipeline of estimating gradients. While the first gradient reflected an overall anterior-posterior axis of connectivity change, the second gradient captured the change in whole-brain INTs. (Pearson’s r = 0.70; see Figure 3 Supplement 1). We therefore conclude that the whole-brain result supports our initially proposed interpretation.

In addition to the addition of Figure 3 Supplement 1, we add the following paragraph to the Results section:

“Although we were mainly interested in the fine-grained functional organization of individual areas, we asked whether the INT-FC relationship holds at the whole-cortex level (see Figure 3 Supplement 1). It is important to note that the whole-cortex estimation reflects overall inter-areal FC differences rather than the fine-grained FC patterns in smaller areas; that is because inter-areal connectivity differences are more pronounced than intra-areal differences. The whole-cortex INT map was highly correlated with the second gradient estimated from the cortical FC matrix (r = 0.70). With these results in hand, we conclude that INTs and FC gradients are closely related and represent a functional principle of organization across the cortex.”

5. There is no strong relationship between within area FC and INT maps in figure 4. Unlike the FC maps that seem to monotonically change along a cortical axis, the INT maps seem to have non-monotonic changes. For example, the INT map for OFC has maximum values both in the anterior-medial and caudal-lateral regions. Similar discrepancies between the INT and FC maps are apparent for PFC. Also, unlike the FC maps, the INT maps seem asymmetrical in the two hemispheres. Unfortunately, the visualization and stats are too dense to probe what is going on. Deeper analyses and explanation are necessary (please see the next comment for a potential solution).

We thank the reviewers for this comment and agree with the assessment. It is important to note that there are crucial differences between the gradient and INT estimation approaches: (1) the gradient estimation pipeline consists of several steps that significantly reduce noise (e.g., the connectivity matrix was made sparse; applying the cosine similarity kernel to build an affinity matrix; and applying a non-linear dimensionality reduction technique to estimate the gradients); (2) the gradients were estimated bilaterally. With respect to the first point, the INT estimation in comparison was done for each voxel individually and no follow-up noise reduction steps have been applied. Previous studies have smoothed the resulting INT maps, but we have decided not to do so since it could erase important functional details. Hence, while the gradients are the result of several processing steps, INTs are a relatively straightforward measure. As a result, the gradient maps appear smooth while the INT maps may appear more variable from voxel to voxel. With respect to the second point, the symmetry of the gradients (although not always present, e.g., see Figure 4C) could be a result of estimating the gradients bilaterally. Other studies (Haak, Marquand, and Beckmann, 2018; Przeździk et al., 2019) have derived the gradients for each hemisphere individually and no major asymmetries have been reported. Nevertheless, this could be a future avenue for further investigating the relationship between INTs and FC gradients. We have decided to estimate the gradients bilaterally because of some of the issues that are inherent with ultrahigh fields. RF transmit field (or B1) inhomogeneities is one of the major drawbacks of high field MRI, even with our high number of transmit channels. The inhomogeneity is a cause for regions of increase and decreased signal intensity in the data. We were worried that these effects might introduce asymmetries between the two hemispheres. We have reason to believe that this issue is partially being taken care of by estimating the gradients bilaterally, but the INTs are still susceptible to these effects. As a result, we believe that the correspondence between INTs and FC gradients might in fact be underestimated in this study. In future studies, we hope to perform full B1 shimming to counteract these shortcomings, but for safety reasons (energy deposition was only modeled for fixed phases of our coil), we did not have that ability before.

For transparency and better interpretation, we added supplementary plots that show that the overall direction of FC and INT change is similar when the values are averaged according to functional subdivisions of the areas under consideration (see Figure 4 Supplement 1; Figure 5 Supplement 1). Additionally, we have worked on the presentation of the INT and FC gradient maps (see the new versions of Figure 3, Figure 4, Figure 4 Supplement 2). We are also adding the following paragraph to the discussion of the manuscript:

“It is noteworthy that there are crucial methodological differences between the gradient and INT estimation approaches: (1) the gradient estimation pipeline consists of multiple steps to reduce the dimensionality of the data, and consequently, the amount of noise (e.g., the FC matrix was made sparse; applying the cosine similarity kernel to build an affinity matrix; and applying a non-linear dimensionality reduction technique to estimate the gradients); (2) the gradients were estimated bilaterally. In comparison to gradients, where the spatial correlation between the voxels’ time series is just the initial step in the pipeline, the INT measure consists of the temporal autocorrelation of individual voxels. No extra steps have been taken for noise reduction. Hence, while gradients are the result of a complex processing pipeline, INTs are a relatively simple calculation. As a result, there is a higher amount of variability in the INT maps which could potentially lead to the underestimation of the FC-INT relationship. With respect to the second point, the symmetry of the gradients could be a byproduct of estimating the gradients bilaterally. Other studies (e.g., Haak et al., 2018; Przeździk et al., 2019) have derived gradients for each hemisphere individually and no major asymmetries have been reported. We estimated the gradients bilaterally because of an issue that is inherent with experimental ultrahigh fields. Specifically, RF transmit field (or B1) inhomogeneities lead to asymmetric signal intensities. We were worried that these effects might introduce asymmetries between the two hemispheres. We have reason to believe that this issue is partially being taken care of by estimating the gradients bilaterally, but the INTs are still susceptible to these effects. This could also result in the underestimation of the FC-INT relationship since there are visible asymmetries in the INT maps.”

6. Many fine level discrepancies are also observed between striatal INTs and cortico-striatal INTs. In the current figures, it is impossible for readers to know if the variations and discrepancies are within the range expected from noise in the data. The alternative is that some of the variations are meaningful but ignored by the authors to keep the narrative simple. Either way, the authors are encouraged to improve their presentation method and analyses in order to provide adequate information for readers. One possibility is to use old-fashioned scatter plots with error bars, where individual data points are different locations within the striatum and the axes are striatal INTs and cortico-striatal INTs. The authors can choose how they'd like to divide striatum into distinct sub-regions (e.g., a simple 3D grid or functional divisions). Similar plots would be useful for figure 4, and more generally anywhere that the authors report a correlation between INTs and FCs.

We thank the reviewer for this comment and refer to our response to Comment 5. To improve interpretability, we added Figure 5 Supplement 1 and the paragraph from Response 5 (this addition to the Discussion is in response to both Comment 5 and 6). Additionally, we have worked on the presentation of the INT and FC gradient maps (see the new versions of Figure 3, Figure 4, Figure 4 Supplement 2).

7. The authors should provide more detailed information about how the hierarchy shown in the x-axis of Figure 1 was computed, so that this can be replicated by other investigators. Some schematic diagram for cortical hierarchy might be also helpful.

We apologize to the reviewers for not having been more explicit in our description. The hierarchies in Figure 1 are based on the studies that have already been cited – i.e., they are hierarchies derived from intrinsic neural timescales estimated from ephys data (Murray et al., 2014; Fascianelli et al., 2019; Cirillo et al., 2018; Cavanagh et al., 2018; Maisson et al., 2021). We added a more extensive description of how we calculated the values for each individual area. Although it could be useful to have a schematic hierarchy, we have refrained from adding one for several reasons: (1) we estimated INTs for both cortical and subcortical structures; (2) to date, there is no established timescale hierarchy at the whole-cortex level (the data is scarce for subcortical structures); (3) Figure 1 was intended as a demonstration of the INT correspondence between ephys and fMRI.

8. The paper would have more value if the same analyses were carried out in other brain areas, such as the temporal cortex, that are not currently covered. If such results are not included, the authors should at least provide a reason why the paper focuses only on the frontal and parietal regions.

We thank the reviewers for their suggestions and agree that further investigations into the relationship between INT and FC gradients (e.g., in the temporal and occipital lobes, or other subcortical structures) are necessary. However, in this paper, we have decided to limit our analyses to the parietal and frontal lobes, and the striatum for two reasons: (1) in the electrophysiology literature, INTs have been almost exclusively investigated within these areas; (2) the expertise of our team is mainly concerned with these parts of the brain.

To address these suggestions, we added the following paragraph to the Discussion of our manuscript:

“We show that INTs are topographically organized along FC gradients throughout the NHP brain. Although we focused our analysis on the frontal and parietal lobes, and the striatum, we also showed that a whole brain analysis recapitulates the same conclusion: resting-state temporal and spatial dynamics are closely related. Future investigations should examine this relationship in other cortical and subcortical brain areas. A promising avenue would be to examine the FC-INT link throughout the brain across different levels of parcellation (and different types of parcellations) While we show that intrinsic timescales are related to FC, other studies have related them to other anatomical and functional markers (Ito, Hearne, and Cole, 2020; Watanabe, Rees, and Masuda, 2019). Hence, further investigations and careful experimentation are needed to understand (1) how temporal dynamics relate to other functional and anatomical characteristics across the brain; (2) their functional relevance and translational potential.”

References:

Autio, J. A., Zhu, Q., Li, X., Glasser, M. F., Schwiedrzik, C. M., Fair, D. A., . . . Vanduffel, W. (2021). Minimal specifications for non-human primate MRI: Challenges in standardizing and harmonizing data collection. Neuroimage, 236, 118082. doi:10.1016/j.neuroimage.2021.118082

Haak, K. V., Marquand, A. F., and Beckmann, C. F. (2018). Connectopic mapping with resting-state fMRI. Neuroimage, 170, 83-94. doi:10.1016/j.neuroimage.2017.06.075

Ito, T., Hearne, L. J., and Cole, M. W. (2020). A cortical hierarchy of localized and distributed processes revealed via dissociation of task activations, connectivity changes, and intrinsic timescales. Neuroimage, 221, 117141.

Jung, B., Taylor, P. A., Seidlitz, J., Sponheim, C., Perkins, P., Ungerleider, L. G., . . . Messinger, A. (2021). A comprehensive macaque fMRI pipeline and hierarchical atlas. Neuroimage, 235, 117997. doi:https://doi.org/10.1016/j.neuroimage.2021.117997

Lagore, R. L., Moeller, S., Zimmermann, J., DelaBarre, L., Radder, J., Grant, A., . . . Adriany, G. (2021). An 8-dipole transceive and 24-loop receive array for non-human primate head imaging at 10.5 T. NMR in Biomedicine, 34(4), e4472.

Logothetis, N. K., Pauls, J., Augath, M., Trinath, T., and Oeltermann, A. (2001). Neurophysiological investigation of the basis of the fMRI signal. nature, 412(6843), 150-157.

Pelekanos, V., Mok, R. M., Joly, O., Ainsworth, M., Kyriazis, D., Kelly, M. G., . . . Kriegeskorte, N. (2020). Rapid event-related, BOLD fMRI, non-human primates (NHP): choose two out of three. Scientific Reports, 10(1), 1-13.

Przeździk, I., Faber, M., Fernández, G., Beckmann, C. F., and Haak, K. V. (2019). The functional organisation of the hippocampus along its long axis is gradual and predicts recollection. Cortex, 119, 324-335. doi:https://doi.org/10.1016/j.cortex.2019.04.015

Shinn, M., Hu, A., Turner, L., Noble, S., Achard, S., Anticevic, A., . . . Bullmore, E. T. (2021). Spatial and temporal autocorrelation weave human brain networks. bioRxiv.

Ugurbil, K. (2016). What is feasible with imaging human brain function and connectivity using functional magnetic resonance imaging. Philosophical Transactions of the Royal Society B: Biological Sciences, 371(1705), 20150361.

Uğurbil, K. (2018). Imaging at ultrahigh magnetic fields: History, challenges, and solutions. Neuroimage, 168, 7-32.

Uğurbil, K., Auerbach, E., Moeller, S., Grant, A., Wu, X., Van de Moortele, P. F., . . . Radder, J. (2019). Brain imaging with improved acceleration and SNR at 7 Tesla obtained with 64channel receive array. Magnetic Resonance in Medicine, 82(1), 495-509.

Watanabe, T., Rees, G., and Masuda, N. (2019). Atypical intrinsic neural timescale in autism. eLife, 8, e42256. doi:10.7554/eLife.42256

Yacoub, E., Grier, M. D., Auerbach, E. J., Lagore, R. L., Harel, N., Adriany, G., . . . Zimmermann, J. (2020). Ultra-high field (10.5 T) resting state fMRI in the macaque. Neuroimage, 223, 117349. doi:https://doi.org/10.1016/j.neuroimage.2020.117349